# m⁶A-mRNA methylation regulates cardiac gene expression and cellular growth

Vivien Kmietczyk[1,2,*] , Eva Riechert[1,2,*] , Laura Kalinski[1,2], Etienne Boileau[1,2,3] , Ellen Malovrh[1,2] , Brandon Malone[1,2,3], Agnieszka Gorska[1,2], Christoph Hofmann[1,2] , Eshita Varma[1,2], Lonny Jürgensen[1,2], Verena Kamuf-Schenk[1,2], Janine Altmüller[4,5], Rewati Tappu[1,2], Martin Busch[1,2] , Patrick Most[1,2], Hugo A Katus[1,2], Christoph Dieterich[1,2,3], Mirko Völkers[1,2]

Conceptually similar to modifications of DNA, mRNAs undergo chemical modifications, which can affect their activity, localization, and stability. The most prevalent internal modification in mRNA is the methylation of adenosine at the N⁶-position (m⁶A). This returns mRNA to a role as a central hub of information within the cell, serving as an information carrier, modifier, and attenuator for many biological processes. Still, the precise role of internal mRNA modifications such as m⁶A in human and murine-dilated cardiac tissue remains unknown. Transcriptome-wide mapping of m⁶A in mRNA allowed us to catalog m⁶A targets in human and murine hearts. Increased m⁶A methylation was found in human cardiomyopathy. Knockdown and overexpression of the m⁶A writer enzyme Mettl3 affected cell size and cellular remodeling both in vitro and in vivo. Our data suggest that mRNA methylation is highly dynamic in cardiomyocytes undergoing stress and that changes in the mRNA methylome regulate translational efficiency by affecting transcript stability. Once elucidated, manipulations of methylation of specific m⁶A sites could be a powerful approach to prevent worsening of cardiac function.

## Introduction

Heart disease, especially heart failure (HF), is a frequent disorder with a considerable burden for our health-care system (Mudd & Kass, 2008; Bui et al, 2011). Although the development of HF depends on variable environmental influences, aberrant changes in myocardial gene expression represent a fundamental feature of both diseased human hearts and many animal models of HF (Tan et al, 2002). Numerous gene expression studies based on profiling of

mRNA abundance in failing human heart tissues or experimental HF models have been performed providing large datasets describing the networks of gene expression control in diseased hearts (Koitabashi & Kass, 2012; Papait et al, 2013; Han et al, 2014). In contrast, only recent research has unveiled the importance of reversible mRNA modifications on gene expression control in different cellular systems (Meyer & Jaffrey, 2014), and previous studies did not capture gene expression control at the level of mRNA methylation in diseased cardiac myocytes. Reversible mRNA modifications have been proposed in 2010 by the He laboratory (He, 2010) and the discovery of fat mass and obesity-associated (Fto) and AlkB homolog 5 RNA demethylase (Alkbh5) proteins as m⁶A demethylases in 2011 has finally shown the dynamic, reversible, and adjustable nature of m⁶A RNA modifications (Jia et al, 2011). In mammals, the writer of m⁶A is a multi-component enzyme consisting of methyltransferase-like 3 (Mettl3), methyltransferase-like 14 (Mettl14), and Wilms tumor 1–associated protein (Wtap) (Liu et al, 2014). In 2012, two groups independently developed methods to assess the in vivo methylation state of m⁶A sites (Dominissini et al, 2012; Meyer et al, 2012), which allowed genome-wide mapping of m⁶A modifications and unveiled the importance of reversible mRNA modifications on gene expression control (Geula et al, 2015; Wang et al, 2015). This work has shown that m⁶A modifications of mRNAs occur and play a crucial role in gene expression control of development, cell growth, metabolism, cellular survival, and intracellular signaling (Fu et al, 2014). The presence of m⁶A across different species; the conservation of writers, erasers, and readers; and the phenotypes associated with changes of m⁶A suggest a fundamental role in physiology and pathophysiology.

Although previous studies did not investigate the role of mRNA modifications in pathological cardiac growth, published reports suggest that m⁶A status regulates cell growth. First and importantly, m⁶A RNA methylation regulates gene expression posttranscriptionally in

[1]Department of Cardiology, Angiology, and Pneumology, University Hospital Heidelberg, University of Heidelberg, Heidelberg, Germany   [2]DZHK (German Centre for Cardiovascular Research), Partner Site Heidelberg/Mannheim, Heidelberg, Germany   [3]Section of Bioinformatics and Systems Cardiology, Department of Cardiology, Angiology, and Pneumology and Klaus Tschira Institute for Integrative Computational Cardiology, University of Heidelberg, Heidelberg, Germany   [4]Cologne Center for Genomics, University Cologne, Cologne, Germany   [5]Center for Molecular Medicine Cologne, University of Cologne, Cologne, Germany

Correspondence: mirko.voelkers@med.uni-heidelberg.de
*Vivien Kmietczyk and Eva Riechert contributed equally to this work as first authors

various cell types (Fu et al, 2014; Wang et al, 2015; Gilbert et al, 2016). Second, enrichment of m⁶A in exons and around the stop codon regions has been shown to regulate translation (Dominissini et al, 2012; Slobodin et al, 2017). Indeed, studies have shown that genetic ablation of m⁶A writers or readers affects translation of specific mRNAs (Fu et al, 2014; Geula et al, 2015; Wang et al, 2015). In addition, loss-of-function mutations in Fto in humans are associated with hypertrophic cardiomyopathies (Boissel et al, 2009). Very recent reports demonstrated the functional importance of the cardiac Fto-dependent m⁶A methylome during myocardial infarction (Mathiyalagan et al, 2018) as well as cardiac growth control by Mettl3 (Dorn et al, 2018). The overall goal of this study was to investigate and elucidate the critical importance of the mRNA epi-modification m⁶A in the heart in gene expression control. We successfully mapped m⁶A transcripts in a murine HF model and in human HF samples. We also delineated the impact of the m⁶A methylase Mettl3 on growth control and defined the gene expression and translational efficiency of identified Mettl3 target genes by translatome analysis with ribosomal-profiling (Ribo-seq).

## Results and Discussion

### Cardiac m⁶A mRNA methylome is dynamic and differs between healthy and diseased human cardiac tissue

Sequencing of m⁶A-specific immunoprecipitated mRNA allows genome-wide mapping of m⁶A modifications. m⁶A dot blots were performed on the m⁶A-positive (IP) and m⁶A-negative (Flow through) fractions and showed successful and specific pull-down of m⁶A-methylated RNA (Fig S1A). To further demonstrate that our protocol selectively enriches for m⁶A-methylated targets, we performed qRT-PCR on the fractions after RNA precipitation. A substantial enrichment of previously published m⁶A-methylated transcripts such as Drd1a was detected in the IP fraction (Meyer et al, 2012). In contrast, transcripts that lack m⁶A enrichment, such as Nde1, were undetectable in the IP fraction (Fig S1B).

Measurements of m⁶A level and sequencing of m⁶A-specific immunoprecipitated mRNA were performed in human failing, dilated cardiomyopathy (DCM) samples and compared with healthy myocardium (control). In line with a recently published report (Mathiyalagan et al, 2018), we found increased m⁶A levels in mRNAs (Fig 1A) isolated from human failing myocardium. We analyzed expression of m⁶A writers and erasers in human heart biopsies from a published DCM patient cohort (n = 33) compared with healthy controls (n = 24) by RNA-seq (Meder et al, 2017). These data did not show any significant changes of Mettl3 or Fto on the transcript level assessed by RNA-seq (Fig 1B). A trend in increased protein levels and RNA levels of Mettl3 could be observed in a small cohort of DCM hearts compared with control samples (Fig S1C), suggesting that only in the subset of DCM patients, Mettl3 expression levels increased.

We identified thousands of genes significantly enriched in the IP fraction from failing myocardium, whereas fewer transcripts were identified in healthy myocardium (Table S1). 1,595 disease-specific methylated transcripts in human DCM samples and only 331 control-specific transcripts were found (Fig 1C). These findings were

validated by qRT-PCR of precipitated mRNA for DCM- or control-specific methylated targets chosen by highest enrichment (Fig S1E). Next, Gene Ontology (GO) term analysis was performed for DCM-specific methylated transcripts. m⁶A-containing transcripts of DCM samples were specifically enriched for genes involved in gene transcription, cell adhesion, and heart development (Fig 1D and E). Conversely, genes with low methylation levels were enriched in processes such as protein targeting and translation. These results indicate that transcripts encoding for transcriptional regulators are highly methylated in DCM hearts. Specifically, m⁶A-containing transcripts were enriched for genes involved in β catenin and calmodulin binding (Fig S1F). Highest methylation status of mRNAs involved in transcription has also been reported in human stem cells (Molinie et al, 2016). Fig 1F shows examples of m⁶A profiles of genes with varying m⁶A levels between non-failing and failing human myocardium. Overall, these data confirm the dynamic and regulatory character of m⁶A in the failing human myocardium.

### m⁶A methylation machinery is present and functional in cardiomyocytes and affects cell growth

Next, the role of mRNA methylation in cardiomyocytes was investigated. Both, m⁶A writer Mettl3 and the eraser Fto are localized in the nucleus of isolated adult cardiomyocytes and neonatal myocytes (NRCM) as previously described for other cell types (Fig S2A) (Gulati et al, 2014; Schöller et al, 2018). We aimed to characterize the consequence of manipulating m⁶A levels in NRCM after siRNA knockdown of Mettl3 or Fto (Fig S2B). Mettl3 knockdown decreased m⁶A levels in myocytes, whereas knockdown of Fto increased overall m⁶A levels (Fig S2C). Consequences of altered m⁶A levels on cell size were analyzed. Hypertrophy of NRCM is blunted by Fto knockdown in response to α-adrenergic stimulation with phenylephrine (PE) treatment, whereas knockdown of Mettl3 increased the cell size (Fig S2D) In line with the augmented cell size, Mettl3 knockdown significantly increased the expression of hypertrophic markers Nppa and Nppb after PE treatment (Fig S2E).

Because our data suggest that lower m⁶A levels are associated with increased cell size, we tested whether increased m⁶A levels would block pathological growth in cardiac myocytes. Fto binds additional RNA species, including small nuclear RNA (snRNA) and tRNA (Wei et al, 2018), and recent reports showed that Fto also demethylates N6,2′-O-dimethyladenosine (m⁶Am) and snRNAs or tRNAs in addition to m⁶A (Mauer et al, 2017; Wei et al, 2018). Thus, we focused on studying Mettl3 in further experiments. Overexpression of the enzymatically active Mettl3 significantly increased m⁶A levels, whereas an enzymatically dead mutant of Mettl3 (Alarcón et al, 2015; Vu et al, 2017) did not increase m⁶A levels compared with the control group (Fig 2A). Correct nuclear localization of both active Mettl3 and inactive Mettl3 was confirmed by immunofluorescence (Fig S2F). Active Mettl3 blocked induction of cell size in response to PE (Fig 2B). In contrast, overexpression of enzymatically inactive Mettl3 mutant did not block cell growth, but slightly augmented cellular size compared with control cells only when treated with PE. Overall, these data show that manipulating m⁶A levels reciprocally regulate myocyte growth response in vitro.

Next, in vivo studies were performed targeting Mettl3 using cardiac-specific, AAV9-mediated Mettl3 overexpression to increase

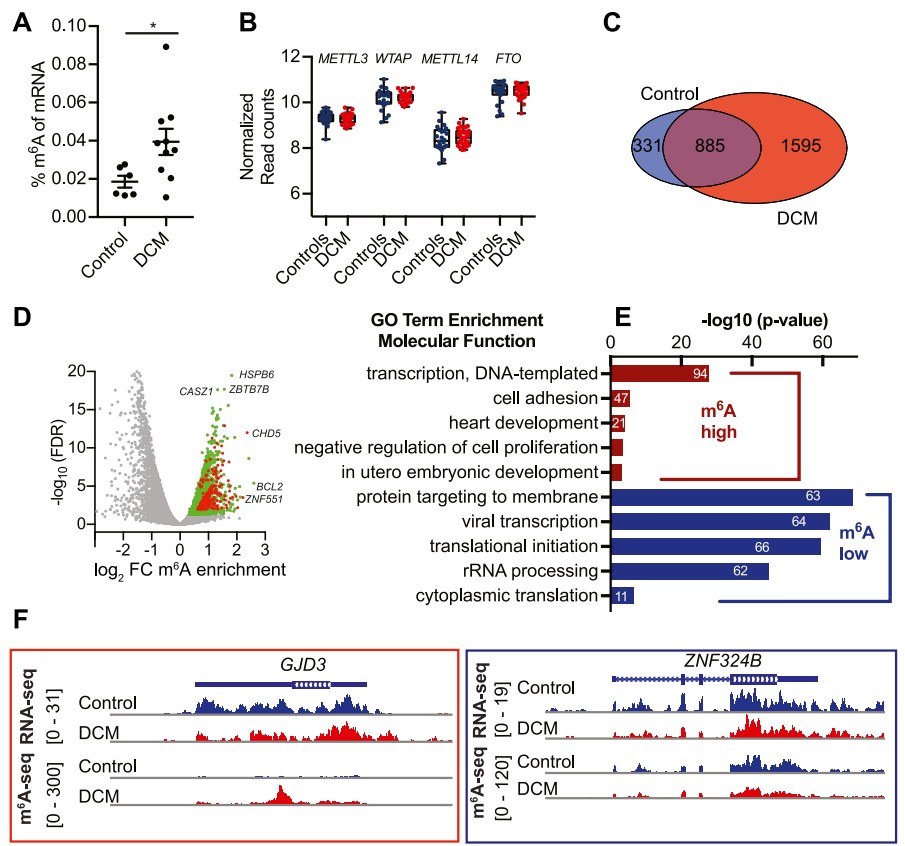

**Figure 1. The m6A mRNA methylome differs between healthy and failing human heart tissue.**
**(A)** Percentage of m⁶A in mRNA from DCM and control samples measured by m⁶A ELISA (n = 6 control and n = 10 DCM, *P = 0.0476 by t test). **(B)** mRNA expression of m⁶A machinery in healthy and DCM cardiac samples analyzed by normalized read counts from RNA-seq from myocardial biopsies (n = 33 DCM and n = 24 control). **(C)** Venn diagram showing overlap of healthy and DCM m⁶A mRNA methylome analyzed by sequencing of m⁶A immunoprecipitated mRNA. **(D)** Gene Ontology (GO) term enrichment analysis for biological processes on the 1,595 DCM-specific methylated mRNAs (high m⁶A) and mRNAs not enriched in IP (low m⁶A). The width of the bars represents the significance (–log₁₀ (adjusted P-value, Fisher's exact test)) of the respective GO term enrichment. **(E)** Volcano plot of sequencing of m⁶A immunoprecipitated mRNA data. Significant enriched transcripts are shown in green. Transcripts involved in transcription (GO term: transcription and DNA-templated are shown in red). Examples of enriched transcripts in DCM hearts are indicated. **(F)** IGV plots of sequencing reads after m⁶A-IP for DCM-specific methylated GJD3 transcript and control-specific ZNF324B transcript.

m⁶A levels. Mettl3-overexpressing mice were subjected to transverse aortic constriction (TAC) surgery to induce pathological hypertrophy. Robust overexpression of Mettl3 and its nuclear localization in myocytes was confirmed (Fig S2G). Overexpression increased m⁶A levels in mRNAs isolated from mice hearts (Fig S2H). Heart size 2 wk after TAC was increased significantly in control mice (Fig 2C–F) and molecular markers of hypertrophy such as *Nppa* and *Nppa* were significantly induced (Fig S2I).

Pathological hypertrophic cellular growth was attenuated in hearts of Mettl3-overexpressing mice, as evidenced by the cross-sectional area of myocytes (Fig 2E and F). Myocytes in Mettl3-overexpressing hearts were significantly enlarged 2 wk post TAC surgery compared with sham-operated animals, but smaller than control TAC mice, without significant differences in hypertrophy marker expression between control TAC and Mettl3 TAC mice (Fig S2I). Consistent with blocked pathological growth, Mettl3 TAC–challenged hearts exhibited decreased fibrosis (Fig 2G and H) and decreased collagen transcription (Fig 2I) when compared with their control TAC-challenged counterparts.

Overall, these data suggest that m⁶A methylation of mRNA directly impacts cardiomyocyte growth, and we provide evidence that mRNA modifications represent a hub for integrating signals which regulate the growth response in the myocardium. Manipulating m⁶A levels in myocytes resulted in altered cellular growth response and cardiac remodeling both in vitro and in vivo. Importantly, phenotypic consequences are dependent on the enzymatic activity of Mettl3. Very recently, an elegant report analyzed the role of Mettl3

during hypertrophic cardiac growth by using cardiac-restricted gain- and loss-of-function mouse models (Dorn et al, 2018). This study showed-in contrast to our study-that increased Mettl3 expression caused (compensated) hypertrophy in vivo, whereas response to TAC was unchanged. Differences in the study design could explain some of the contrasting data. Dorn et al, 2018 used a transgenic model in the FVB background, where we used a C57Bl6/N background for the in vivo studies. Moreover, we used an AAV-based approach to overexpress Mettl3, which resulted in lower overexpression levels than a transgenic approach driven by the alpha myosin heavy chain (aMHC)-promoter in the study by Dorn et al, 2018. However, both studies point towards an important role of epitranscriptomic control on cell growth. More studies are clearly needed to fully understand this novel stress–response mechanism in the heart for maintaining normal cardiac function.

## m⁶A regulates RNA stability and translation efficiency in myocytes

Our loss-/gain-of-function experiments show that altered expression of Mettl3 regulates cell size both in vitro and in vivo. One possible mechanism could involve a regulatory role of m⁶A in translational control of specific mRNAs, particularly under cellular stress conditions. Although mRNA translation is a complex and highly coordinated process, neither the importance nor the mechanisms of translational regulation in myocytes are studied to the same extent as for transcriptional control.

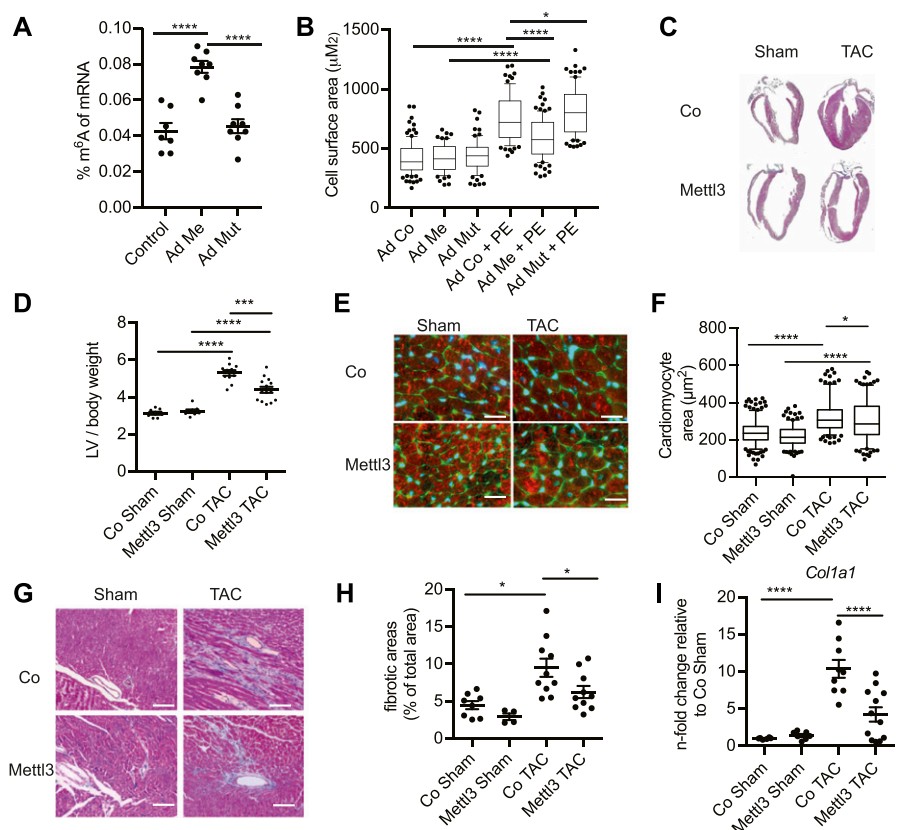

**Figure 2. m⁶A affects cell growth and cardiac function.**

**(A)** Percentage of $m^6A$ in mRNA in NRVMs after Mettl3 or Mettl3 enzymatically inactive mutant overexpression measured by $m^6A$ ELISA (n = 7 for control and n = 8 for Mettl3 WT and Mettl3 mutant). ****$P < 0.0001$ by one-way ANOVA followed by Bonferroni's post hoc comparisons. **(B)** Cell surface of NRVMs overexpressing Mettl3 or inactive mutant compared with control virus with and without PE stimulation (n = 3 independent biological experiments with >69 cells for each group *$P < 0.0047$, ****$P < 0.0001$ by one-way ANOVA). **(C)** Gross morphology of either control virus or Mettl3-injected mouse hearts after either sham or TAC surgery in 10-wk-old mice. **(D)** Ratio of LV weight to tibia length of control and Mettl3 OE mice (n = 10–12, ***$P < 0.001$, ****$P < 0.0001$ by one-way ANOVA). **(E)** Immunofluorescence of heart sections WGA (green), sarcomeric actin (red), and nuclei blue. Bar graph: 20 $\mu M$. **(F)** Cell surface area measurement from WGA staining (n = 4 animals per group and 200–300 cells in total, *$P < 0.05$, ****$P < 0.0001$ by one-way ANOVA). **(G)** Masson trichrome staining on paraffin heart sections. Bar graph: 80 $\mu M$. **(H)** Quantification of fibrotic area from Masson trichrome staining (n = 4–10, *$P < 0.05$ by one-way ANOVA). **(I)** mRNA expression of *Col1a1* analyzed by qPCR (n = 6–14, ****$P < 0.0001$ by one-way ANOVA).

To understand the mechanism by which Mettl3 regulates cell size, we first defined the mRNA methylome early after TAC. Sequencing of $m^6A$-specific immunoprecipitated mRNA was performed 2 d after surgery. This time point was chosen based on previous studies and defines a transition between early adaptation to compensated tissue remodeling with robust changes in gene expression and beginning of pathological growth (Volkers et al, 2013).

Surprisingly, the percentage of $m^6A$ in mRNA decreased substantially from 0.026% in sham-operated animals to 0.012% in response to a TAC surgery (Fig S3A). Also, fewer transcripts were enriched after sequencing of $m^6A$-specific immunoprecipitated mRNAs compared with sham surgery (Fig S3B and Table S2). In total 1,567 genes were enriched after our IP protocol from sham-operated mice. This was associated with a decrease in Mettl3 expression 2 d post TAC surgery (Fig S3C). In line with our human data, $m^6A$-containing RNAs are involved in transcriptional regulation and signal transduction (Fig S3D).

Next, we analyzed the impact of mRNA methylation on translational efficiency in vivo.

To investigate translational control in myocytes, we used the RiboTag system (Sanz et al, 2009) and ribosomal sequencing (Ingolia, 2016) (Ribo-seq) to purify and identify actively translating transcripts from cardiac myocytes after TAC surgery. Translational regulation was assessed by Ribo-seq in combination with RNA-seq in mice 2 d after TAC or sham surgery, when overall $m^6A$ levels were changed (Fig S3A). We followed methylated transcripts in our Ribo-seq data (Fig 3A). Transcripts that were still methylated in TAC-operated mice were highly translated in response to TAC surgery (Fig 3A and B), whereas transcript levels measured by RNA-seq were unchanged (Figs 3B and S3E), suggesting that $m^6A$ affects translational efficiency during pathological growth. Again, highly translated methylated transcripts in TAC-operated mice were enriched for transcriptional regulation and cardiac muscular proteins, indicating a shift of the mRNA methylation towards transcripts of cardiomyocyte function and growth (Fig S3F).

Finally, we aimed to determine the Mettl3-dependent translatome in cardiac myocytes.

Because Ribo-seq methods still require a large number of cells, we used the murine HL-1 cardiomyocyte cell line as our source material. HL-1 cells can be used as a cardiomyocyte model as they have key characteristics of cardiac myocytes, although their metabolism and structure are less organized than primary cardiac myocytes (Claycomb et al, 1998; Eimre et al, 2008). Confluent and spontaneously beating murine HL-1 cells were infected with Mettl3 adenovirus to increase $m^6A$ levels (Fig S3G). We identified Mettl3-dependent, highly differentially translated mRNAs by Ribo-seq (Figs 3C, S3H, and Table S3). Mettl3 targets were involved in regulation of cell cycle, DNA damage response, and again transcriptional regulation (Fig 3D). We followed these differentially expressed transcripts in our in vivo Ribo-seq data from TAC-operated mice. Transcripts that are highly translated in Mettl3-overexpressing cardiac myocytes were highly translated in response to TAC surgery and vice versa (Fig 3E). In contrast, transcript levels of these

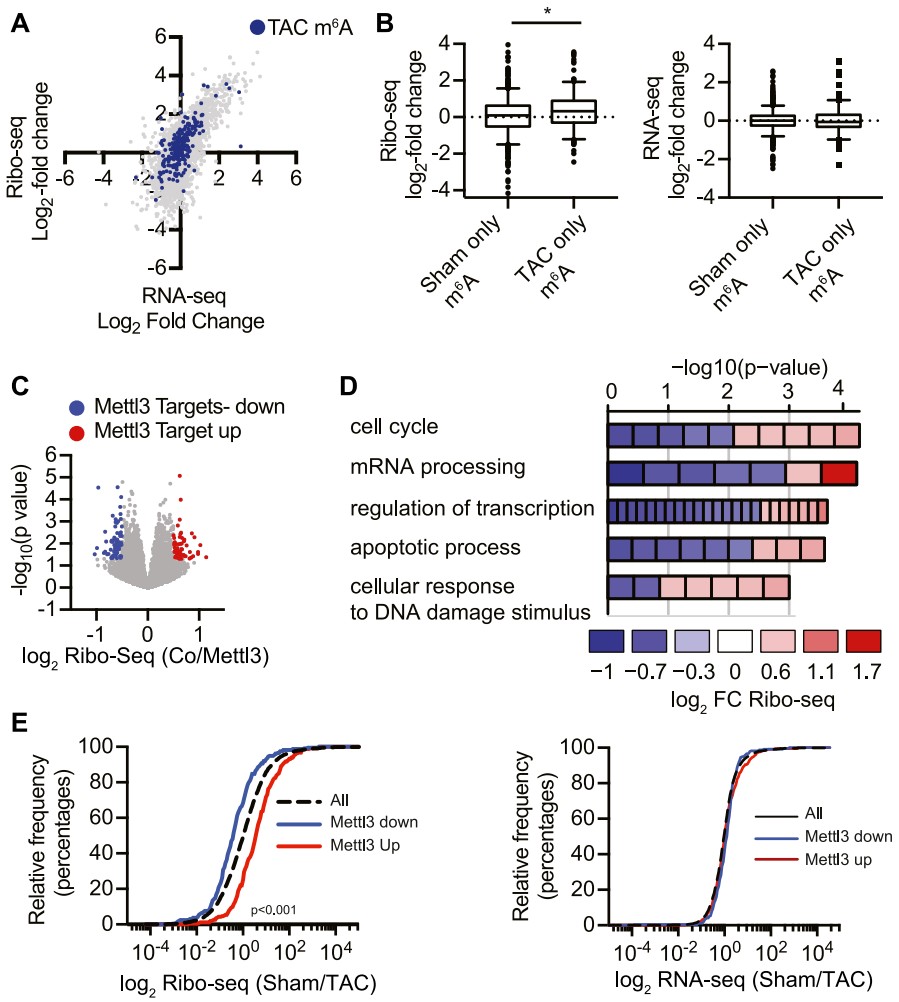

**Figure 3.  m⁶A affects translation efficiency.**
**(A)** Scatter blot of fold change of RNA-seq and Ribo-seq (sham/TAC) from murine hearts 2 d after TAC surgery. All transcripts (gray), enriched in m⁶A-IP (blue). **(B)** Box plots of fold change (sham/TAC) of Ribo-seq and RNA-seq from murine hearts 2 d after TAC surgery for methylated transcripts in sham-operated mice (sham only m⁶A) or post TAC surgery (TAC only m⁶A). *P < 0.05 by t test. **(C)** Volcano blot of Ribo-seq data from HL-1 cells overexpressing Mettl3. **(D)** GO term enrichment analysis (biological process) for Mettl3 target mRNAs, which are identified as regulated in Ribo-seq. The width of the bars represents the significance (−log₁₀ (adjusted P-value, Fisher's exact test)) of the respective GO term enrichment. Colors depict the log fold change (logFC) of individual genes within the GO category. **(E)** Cumulative fraction of mRNAs relative to their fold change of Ribo-seq and RNA-seq (sham/TAC) between all transcripts and Mettl3-regulated transcripts 2 d after TAC surgery (Kolmogorov–Smirnov test, P < 0.001).

methylated transcripts genes were again unchanged in response to TAC, indicating that mRNA methylation by Mettl3 affects translational efficiency of specific transcripts during pathological growth in cardiac myocytes.

We validated Mettl3-dependent translation of two candidates (*Arhgef3* and *Myl2*) by qRT-PCR on polysomal fractions from control or Mettl3-overexpressing cardiomyocytes. Overall translation was unchanged in Mettl3-overexpressing myocytes as assessed by polysome profiles (Fig 4A). However, *Arhgef3* was significantly less, whereas *Myl2* translated more in our HL-1 Ribo-seq data. In contrast, mRNA levels remain unchanged after Mettl3 overexpression (Fig 4B). As predicted by our Ribo-seq data, *Arhgef3* transcript levels were decreased in polysomal fractions, whereas *Myl2* levels increased after Mettl3 overexpression (Fig 4C). Increased Mettl3 dependent methylation of *Arhgef3* and *Myl2* transcripts were also validated by qRT-PCR after m⁶A-precipitated mRNA from Mettl3- or control-HL-1 cells compared with the input mRNA (Fig 4D). Previous studies suggested that methylation of transcripts affects mRNA stability and decay (Wang et al, 2015). In line, mRNA half-life measurement by blocking transcription with actinomycin D in NRCM showed that Mettl3 decreased the mRNA stability of the

translationally down-regulated *Arhgef3* or *Polr3d* (Fig 4E), whereas stability of *Myl2* mRNAs was increased by Mettl3. Finally, Mettl3 overexpression caused decreased Arhgef3 protein levels, whereas Myl2 levels were increased in Mettl3-overexpressing myocytes (Fig 4F and G). In contrast to our in vitro data, mice overexpressing Mettl3 in the heart showed decreased Myl2 expression both during sham and TAC conditions (Fig 4H and I), whereas Arhgef3 expression increased both during TAC conditions in Mettl3-overexpressing mice and in control animals. Intriguingly, m⁶A enrichment of Arhgef3 and Myl2 changed at different time points after TAC compared with sham-operated animals (Fig S3I). We speculate that longer Mettl3 overexpression results in different methylation patterns in vivo compared with our in vitro data. Moreover, different expression levels of m⁶A reader proteins could certainly affect translation of methylated transcripts after long-term overexpression of Mettl3. Ultimately, more studies are needed to fully understand how m⁶A methylation regulates translation in myocytes.

In summary, our data suggest that m⁶A regulates and affects cardiomyocyte fate by adding a posttranscriptional regulation step to gene expression by influencing mRNA stability and translation efficiency. Using Ribo-seq, we identified subsets of mRNAs, which

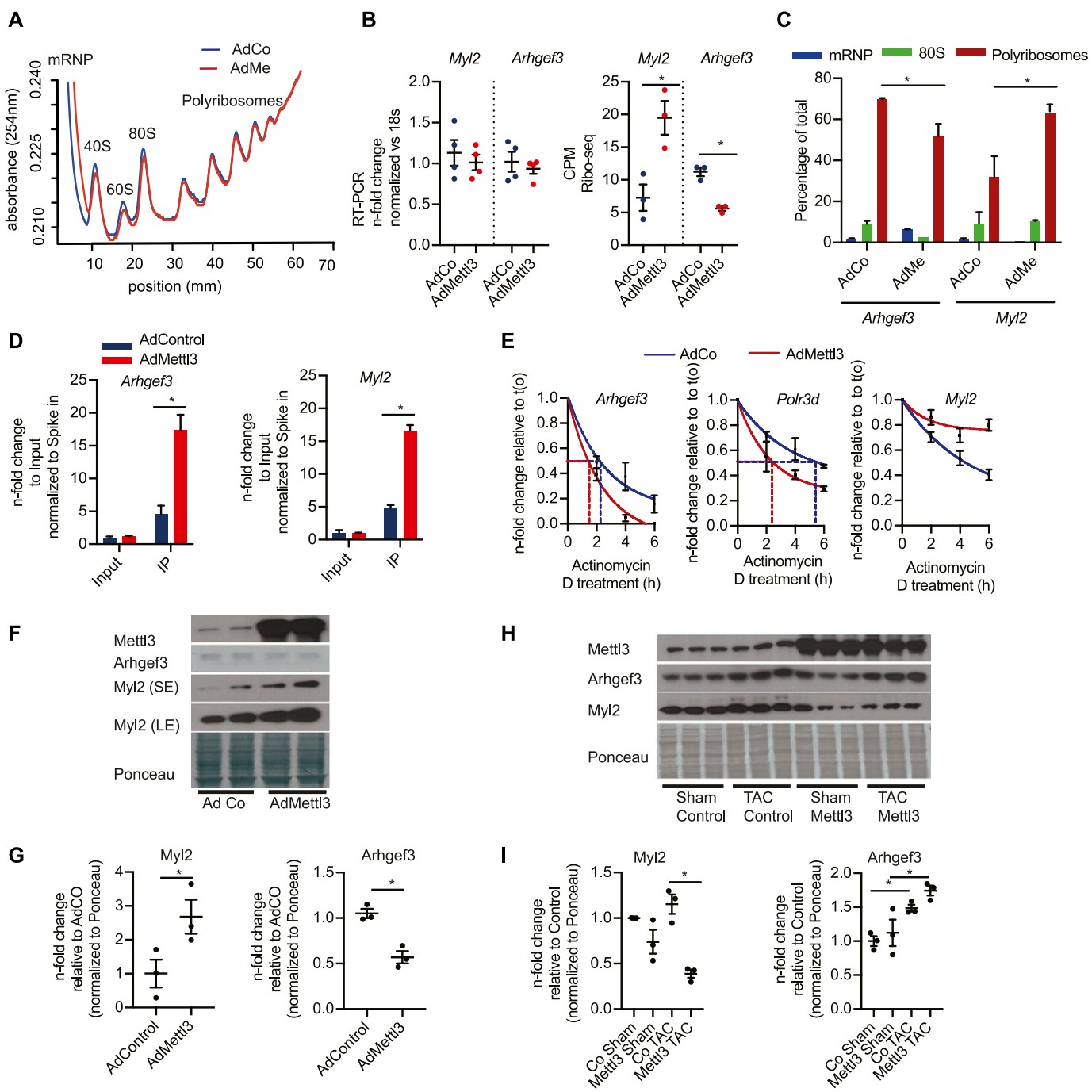

**Figure 4. m⁶A affects expression of target genes by regulating transcript stability.**

**(A)** Polysome profile of control and Mettl3-overexpressing cardiomyocytes (HL-1 cells). **(B)** RT-PCR (left panel) and Ribo-seq counts (right panel) for *Myl2* and *Arhgef3* in HL-1 cells (*$P < 0.05$ after EdgeR analysis for the Ribo-seq quantification, n = 3). **(C)** Percentage of total transcript abundance of *Arhgef3* and *Myl2* in mRNPs, 80s, and polyribosomes (*$P < 0.05$ by t test; n = 2 independent experiments). **(D)** Enrichment of m⁶A-mRNA after m⁶A-IP shown by qPCR of *Arhgef3* and *Myl2* after Mettl3 overexpression. *$P < 0.01$ by t test; n = 3 independent experiments. **(E)** mRNA stability of Mettl3 targets *Arhgef3*, *Polr3d*, and *Myl2* analyzed by qPCR upon actinomycin D treatment. **(F, G)** Immunoblots and (G) quantification for Arhgef3 and Myl2 in HL-1 cells after confirming the Mettl3 overexpression. $P < 0.05$ by t test; n = 3 independent experiments. **(H, I)** Immunoblots and (I) quantification for Arhgef3 and Myl2 in mice after sham or TAC surgery cells after Mettl3 overexpression. *$P < 0.05$ by one-way ANOVA (n = 3 for each group; SE, short exposure; LE, long exposure).

are translationally dependent on methylation status or Mettl3 activity. Our findings support the paradigm that mRNA modifications in the heart play important roles in the heart by 1) validating

the expression of Mettl3 and Fto in the myocardium, 2) demonstrating dynamic mRNA methylome in HF both in human and murine hearts, 3) showing effects upon cell growth in vitro and in

vivo, and 4) providing evidence that mRNA methylation also influences mRNA stability and translation in the heart. We also identified Mettl3 as a key cardiac methylase that regulates cardiac m[6]A and growth control.

However, the exact mechanisms how Mettl3 regulates cellular growth control and which specific targets are responsible for the phenotypic consequences are unknown. Overall methylation early after pathological stress in murine hearts is decreased, whereas human HF samples showed increased m[6]A levels. Whether this is due to altered Mettl3 or Fto activity needs to be investigated. Also, it is unknown how enzymatic activity of writers and erasers is regulated. Similarly, how increased m[6]A levels are explained in a complex interplay of writers and erasers in failing myocardium remains unclear. Recently, a beneficial role of the demethylase Fto in cardiac function after myocardial infarction has been reported (Mathiyalagan et al, 2018). This study identified that Fto selectively demethylates cardiac contractile transcripts such as *Serca2*a or *RyR2* and thereby increases calcium handling and cardiac contractility. Our study did not identify a specific effect of Mettl3 on the translational status of *Serca2*a or *RyR2*. Whether writers or erasers compete with targets under different stress conditions and how this might be regulated is unknown and requires further studies. Our data suggest that the enzymatic activity of Mettl3 positively and negatively regulates translational status of specific transcripts. This clearly could be the result of regulation of mRNA stability, degradation, or direct rate of translation. In fact, m[6]A has been shown to have both stimulatory (Meyer et al, 2015; Wang et al, 2015) and inhibitory effects (Choi et al, 2016; Slobodin et al, 2017) on translation. We speculate that Mettl3 affects translational efficiency by methylating mRNAs encoding for proteins involved in transcriptional regulation which could fine-tune the response to cellular stress. Alternatively, because m[6]A reader such as Ythdf1-3 (Wang et al, 2015; Shi et al, 2018, 2017) can regulate translational efficiency of methylated transcripts, especially in cellular stress conditions, changes in expression or activity of those readers might alter translational efficiency of methylated transcripts. We validated Mettl3-dependent gene expression regulation of two interesting candidates in follow-up experiments. Mettl3 decreases Arhgef3 protein levels in vitro. Arhgef3 (also known as Xpln) is a Rho guanine nucleotide exchange factor and has been found to interact with the protein kinase mechanistic target of rapamycin (mTOR) (Khanna et al, 2013). Increased levels of Arhgef3 have been shown to stimulate mTORC1. Moreover, increased activity of mTORC1 has been shown to contribute to cardiac hypertrophy and HF (Volkers et al, 2013; Sciarretta et al, 2018), and ongoing studies will investigate the role of Arhgef3 on mTORC1 regulation during pathological growth. Myosin light chain-2 (Myl2) expression increased after Mettl3 overexpression in vitro, and its expression is slightly increased after TAC surgery. Myl2 is a sarcomeric protein that belongs to the EF-hand calcium-binding protein superfamily. Genetic loss-of-function studies in mice demonstrated the essential role for Myl2 in cardiac contractile function (Sheikh et al, 2015), but it is unknown if increased expression of Myl2 is needed in addition to the characterized regulation by phosphorylation for the adaptation to increased workload in response to acute pressure overload. Additional studies will be needed to fully understand whether m[6]A-dependent gene expression control of Arhgef3 and Myl2 levels

causally contributes to inhibition of pathological growth after Mettl3 overexpression.

Thus, mRNA modifications represent an additional way of controlling gene expression in the myocardium. Once specific targets will be elucidated, manipulations of m[6]A levels could be a powerful approach to prevent worsening of cardiac function. It may become possible to alter mRNA modifications therapeutically by engineering RNA-modifying enzymes with altered substrate specificity or to test whether novel small molecules that affect the formation of mRNA modifications have therapeutic values. Those approaches will open exciting prospects of developing new therapies for diseases caused by m[6]A dysregulations, including heart diseases.

# Materials and Methods

### Animal tissues

All experiments were performed in 10-wk-old male mice unless otherwise indicated. The RiboTag mice were purchased from the Jackson Laboratory (JAX ID 011029). The mice were housed in a temperature- and humidity-controlled facility with a 12-h light–dark cycle. The RiboTag mouse was bred to the αMHC-Cre mice to obtain *Rpl22*[HA]-expressing homozygous mice in cardiac myocytes. At 10 wk of age, male mice underwent TAC (27 gauge needle) or sham operation, as previously described (Rockman et al, 1991) (Doroudgar et al, 2015). For echocardiography, the mice were anesthetized with 2% isoflurane and scanned using a Vevo2100 imaging system (Visual Sonics) as previously described (Völkers et al, 2014). Institutional Animal Care and Use Committee approval was obtained for all animal studies.

### Human tissue

The characterization of samples and patient data has been approved by the ethics committee, and medical faculty of Heidelberg and participants have given written informed consent. Biopsy specimens were obtained from the apical part of the free left ventricular wall (LV) from DCM, or control LV biopsy specimens were obtained from stable and symptom-free patients after heart transplantation. Biopsies were rinsed with NaCl (0.9%) and immediately transferred and stored in liquid nitrogen until DNA or RNA was extracted. Total RNA was extracted from biopsies using the RNeasy kit according to the manufacturer's protocol (QIAGEN). For m[6]A-seq analysis, RNA has been isolated from explanted human hearts from patients suffering chronic HF. Control RNA from non-failing LV tissue has been obtained commercially (BioCat) or from donor hearts that could not be used for transplantations.

### Preparation of tissue lysates

For heart homogenates, male αMHC-Cre:RiboTag mice were euthanized, and their hearts were quickly excised, washed in PBS containing 100 μg/ml cycloheximide (CHX), and snap-frozen in liquid nitrogen as previously described (Siede et al, 2017). LV tissue was homogenized using a tissue homogenizer (Bullet Blender, Next

Advance) in five volumes of ice-cold polysome buffer (20 mM Tris, pH 7.4, 10 mM MgCl, 200 mM KCl, 2 mM DTT, 100 μg/ml CHX, 1% Triton X-100, and 1 U DNAse/μl) containing 100 μg/ml CHX. Homogenates were centrifuged at 4°C and 15,000 g for 10 min, and the supernatant was immediately used in the further steps. 100 μl of lysate was used as input, from which RNA was extracted using Trizol. The remaining lysate was used for cell type specificity analysis of translated RNAs. This was achieved by anti-HA IP of polysomes. Anti-HA magnetic beads (88836; Thermo Fisher Scientific; 100 μl per heart) were washed with 1,000 μl polysome lysis buffer three times. The lysate was then added to anti-HA magnetic beads and incubated with rotation at 4°C overnight. The beads were then washed three times with 500 μl of high salt buffer (20 mM, Tris pH 7.4, 10 mM MgCl, 300 mM KCl, 2 mM DTT, and 1% Triton X-100). The washed beads were subjected to RNA extraction for library construction and subsequent high-throughput sequencing using the Illumina TruSeq Ribo Profile kit or immunoblotting analysis.

## Polysome profiling

Sucrose solutions were prepared in polysome gradient buffer and 20 U/ml SUPERase-In (Ambion). Sucrose density gradients (10–50% wt/vol) were freshly made in SW40 ultracentrifuge tube using a BioComp Gradient Master (BioComp) according to the manufacturer's instructions. Tissue lysates were loaded onto sucrose gradients, followed by centrifugation for 250 min at 220,000 g, 4°C, in an SW40 rotor. Separated samples were fractionated at 0.375 ml/min by using a fractionation system BioComp Gradient Station (BioComp) that continually monitors OD254 values. The fractions were collected into tubes at 0.3-mm intervals. Isolation of RNA from fractions was performed by QIAzol (QIAGEN) and Quick RNA Miniprep kit (Zymo Research) followed by cDNA synthesis of 200 ng RNA using the iScript cDNA Synthesis kit (Bio-Rad). Before extraction, 20 pg of Renilla luciferase in vitro–transcribed mRNA were added to the fractions as an internal standard. qPCRs were run with 1:10 dilutions of the cDNA samples. Data were normalized to the luciferase and the sample input signal.

## m⁶A immunoprecipitation and sequencing

Human and mouse heart tissue were lysed in 700 μl lysis buffer (20 mM Tris, pH 7.4, 10 mM MgCl, 200 mM KCl, 2 mM DTT, 100 μg/ml CHX, 1% Triton X-100, and 1U DNAse/ul) using a tissue homogenizer (Bullet Blender, Next Advance). The tissue was homogenized further by passing the lysate through a 23-gauge syringe needle 10 times. Homogenates were centrifuged at 4°C and 18,000 g for 10 min, and the supernatant was immediately used in the further steps. For complete lysis, the samples were kept on ice for 10 min and subsequently centrifuged at 20,000 g to precipitate cell debris.

Ribosomal RNA was depleted from total RNA samples by ribo zero (MRZH11124; Illumina). 50 μg of total RNA was used. Non-IP RNA was stored for RNA-seq analysis.

Ribosomal depleted RNA was subjected to m⁶A IP. mRNAs were immunoprecipitated by m⁶A antibody (ab151230; Abcam) precoupled protein A/G magnetic beads (88802; Pierce). Elution was performed in all cases with m⁶A salt. Libraries were constructed using Illumina TruSeq protocols and sequenced by the Cologne

Center as 75-bp single reads. All sequencing data were subjected to the same preprocessing steps: reads are trimmed (adapter removal) and quality clipped with FlexBar (Dodt et al, 2012). All remaining reads (>18 bp in length) are mapped either against the murine 45S rRNA precursor sequence (BK000964.3) or the human 45S rRNA precursor sequence (NR_046235.1) with Bowtie 2 to remove rRNA contaminant reads. We used the splice-aware STAR read aligner (Dobin et al, 2013) (release 2.5.1b) to map reads from input RNA and m⁶A IP libraries against the respective EnsEMBL genome assemblies (release 87). Differential enrichment analyses on IP/input gene read counts was conducted using the edgeR package (McCarthy et al, 2012) for each condition separately. We used a two-factor design (IP or input, sample number) and assessed the significance of the IP/input contrast. The adjusted P-values were reported for every expressed gene locus (>10 counts over all input samples). Enrichment was calculated as log2-fold change (logFC) of ratio of reads from immunoprecipitated mRNA sequencing and mRNA sequencing and defined as enriched if logFC > 0 and FFDR < 0.05 and as not enriched/m⁶A low if logFC < −1 and FDR < 0.05.

## Parallel generation of ribosome profiling and RNA libraries

To accurately dissect translation and transcription, both Ribo-seq and RNA-seq libraries were prepared for each biological replicate from aMHC-Cre–positive RiboTag mice after sham or TAC surgeries. Cardiomyocyte-specific ribosomes were isolated like previously described (Siede et al, 2017). Briefly, the tissue was homogenized in polysome buffer, and lysate was used as the input for RNA-seq.

The remaining lysate was used for direct IP of polysomes. Anti-HA magnetic beads (88836; Thermo Fisher Scientific; 100 μl per heart) were washed with 1,000 μl polysome lysis buffer three times. The lysate was then added to HA magnetic beads and incubated with rotation at 4°C overnight. The beads were then washed three times with 500 μl of high salt buffer (20 mM, Tris pH 7.4, 10 mM MgCl, 300 mM KCl, 2 mM DTT, and 1% Triton X). The washed beads were subjected to RNA extraction for Ribo-seq library construction. Libraries were generated according the to the mammalian ARTseq kit (Illumina). Barcodes were used to perform multiplex sequencing and create sequencing pools containing at least eight different samples and always an equal amount of both RNA and RPF libraries. Sample pools were sequenced on the NextSeq platform using 75-SE sequencing chemistry.

## Sequencing data processing and quality control

Ensembl annotations were used for the analysis. For each gene, all its annotated coding sequnce (CDS) regions were extracted and then concatenate into a single *metagene*. A previously published pipeline was used for quality control and data processing (Malone et al, 2017). A fully Bayesian translation prediction approach was used—RP-BP. Standard techniques were used to construct the profiles. Adapters and low-quality reads were removed. Reads mapping to ribosomal and tRNA sequences were removed. Reads to the genome were aligned with a splice-aware aligner and reads removed with multiple genomic alignments. The Ribo-seq profiles are constructed using the same steps; however, the BPPS method was used to shift aligned reads to properly account for the P-site of

the ribosome (Malone et al, 2017). Notably, this ensures alignments at the start codon of the transcript are properly included in the Ribo-seq profile. For the analyses, we follow the protocol as outlined in the study by Schafer et al (2015). We pinpoint translational control by identifying read counts in Ribo-seq data across conditions and categorize differential gene expression events. We use the edgeR package (McCarthy et al, 2012) with a four-factor design matrix (RNA-seq cond1, RNA-seq cond2, Ribo-seq cond1, and Ribo-seq cond2) to accomplish this task. We only consider data points with read count observations across all replicates. The Ribo-seq log2-fold changes are counts per million reads in TAC versus sham-operated mice.

### Primary cell culture

Neonatal rat ventricular myocytes (NRVM) were isolated by enzymatic digestion of 1–4-d-old neonatal rat hearts and purified by Percoll density gradient centrifugation before plating, as described (Doroudgar et al, 2015). Cardiac myocytes were cultured at various densities, as described in the figure legends. Tissue culture plates were coated with 5 μg/ml fibronectin in serum-free DMEM/F-12 medium for 1 h. Briefly, $5 \times 10^5$ Percoll-purified cardiac myocytes were plated on 35-mm fibronectin-coated plastic culture wells in DMEM/F-12 medium containing 10% fetal bovine serum. After 24 h, the medium was replaced with DMEM/F-12 medium containing 2% fetal bovine serum, and the cultures were maintained for 24 h before treatment.

Cells were treated with 100 μM PE for 24 h for cell size analysis. Raw data about cell size are shown in Table S4. Adult LV cardiomyocytes were obtained using a collagenase digestion method as described in detail elsewhere (Most et al, 2004).

### HL-1 cell culture

HL-1 cardiomyocytes were maintained as described (Claycomb et al, 1998).

### AAV9 generation and systemic in vivo AAV9 cardiac-targeted gene transfer protocol

In vivo cardiac-targeted Mettl3 expression in normal mouse hearts was obtained using tail vein injection of an AAV9 harboring the Mettl3 gene driven by a cardiomyocyte-specific CMV–MLC2v0.8 promoter as previously described (Völkers et al, 2013).

### Generation of the mutant Mettl3 (Mettl3 Mut) construct

Two point mutations were introduced within the catalytic site of Mettl3 (Vu et al, 2017) in two mutagenesis steps by using the following PCR protocol with pAd_Mettl3 as a parent plasmid and specifically designed primers containing the desired mutations using the Agilent QuikChange II Site-Directed Mutagenesis kit as per the manufacture's instruction. The obtained plasmid was transformed into NEB α-5 E. coli following the High Efficiency Transformation Protocol from New England Biolabs. Validation of the created plasmid was performed by sequencing, and the mutant construct was cloned into a pAd vector using the LR recombination protocol by Thermo Fisher Scientific.

### Immunocytofluorescence of mouse heart sections

Hearts were cleared by retroperfusion in situ with PBS at 70 mmHg, arrested in diastole with 60 mM KCl, fixed by perfusion for 15 min with 10% formalin (HT501128; Sigma-Aldrich), excised, fixed in formalin for 24 h at room temperature, and embedded in paraffin. Paraffin-embedded hearts were sectioned and placed on slides, which were then deparaffinized and then rehydrated. Antigen retrieval was achieved by boiling the slides in 10 mM citrate (pH 6.0) for 12 min, after which the slides were washed several times with distilled water, and once with Tris/NaCl, or TN buffer (100 mM Tris and 150 mM NaCl). Primary antibodies were diluted in TNB and added to slides which were incubated at 4°C for ~12–16 h. The samples were then washed with TN buffer and incubated with secondary antibodies at room temperature in the dark for 2 h. Images were obtained using a Zeiss Observer.Z1 fluorescence microscope. Images were obtained with a 20× objective.

### Immunoblotting

Samples were combined with the appropriately concentrated form of Laemmli sample buffer and then boiled before SDS–PAGE followed by transfer to polyvinylidene difluoride (PVDF) membranes.

### Quantitative real-time PCR

Total RNA was isolated from frozen heart or cultured cells by using Quick-RNA MiniPrep (Zymo Research) and reverse-transcribed into cDNA by using iScript Reverse Transcription Supermix (Bio-Rad). Quantitative real-time PCR was performed on all samples in triplicate using iTAQ SYBR Green PCR kit (Bio-Rad) according to the manufacturer's instructions. Information about the primers used in the study is shown in Table S5.

### m$^6$A ELISA

To detect overall levels of m$^6$A displaying RNAs, the m$^6$A ELISA (ab185912; Abcam) was performed per the protocol. Total RNA was isolated from the cultured cell lysates by using the Quick-RNA MiniPrep kit form Zymo Research including an on-gDNA removal step and an additional in-column DNAse I digest step. mRNA was isolated by polyA enrichment using oligo(dt) magnetic beads (NEB) according to the manufacturer's instructions.

### m$^6$A dot blotting

mRNA was denatured at 75°C for 5 min, spotted, and cross-linked to a positively charged nylon membrane 2× in UV-Stratalinker with 1,800 μJ/cm$^2$ at 254 nm. The membrane was probed with m$^6$A antibody (ab151230, 1:1,000; Abcam).

### GO analysis

GO term enrichment analysis was performed using the subset of expressed protein-coding genes as background set (DAVID). Categories with a P-value <0.05 were retained, and a subset was visualized with R

studio package CellPlot showing enrichment *P*-value, expression fold changes, and number of genes simultaneously.

## Data availability

The datasets produced in this study are available in the following databases: $M^6A$-seq, RNA, and Ribo-seq data have been uploaded to SRA with the following ID: SRP156230.

## Statistics

Cell culture experiments were performed at least two to four times with n = at least two biological replicates (cultures) for each treatment. In vivo experiments were performed on at least three biological replicates (mice) for each treatment. The investigators have been blinded to the sample group allocation during the experiment and analysis of the experimental outcome. Unless otherwise stated, values shown are mean ± SEM and statistical treatments are one-way ANOVA followed by Bonferroni's post hoc comparisons.

## Supplementary Information

## Acknowledgements

A Gorska, P Most, HA Katus, M Völkers, and C Dieterich acknowledge the DZHK (German Centre for Cardiovascular Research) Partner Site Heidelberg/Mannheim. C Dieterich acknowledges funding from the Klaus-Tschira Stiftung GmbH. M Völkers is supported by Deutsche Forschungsgemeinschaft (DFG VO 1659 2/1 and DFG VO 1659 4/1), and Baden Württemberg Stiftung and E Malovrh by Heidelberg Biosciences International Graduate School.

## Author Contributions

V Kmietczyk: data curation, formal analysis, investigation, methodology, and writing—original draft, review, and editing.
E Riechert: data curation, formal analysis, investigation, methodology, and writing—original draft, review, and editing.
L Kalinski: formal analysis, investigation, and methodology.
E Boileau: resources and formal analysis.
E Malovrh: formal analysis, investigation, and methodology.
B Malone: software and methodology.
A Gorska: supervision, investigation, methodology, and writing—review and editing.
C Hofmann: formal analysis and investigation.
E Varma: formal analysis and investigation.
L Jürgensen: investigation and methodology.
V Kamuf-Schenk: investigation and methodology.
J Altmüller: resources, formal analysis, and methodology.
R Tappu: formal analysis and methodology.
M Busch: formal analysis and investigation.
P Most: methodology.
HA Katus: conceptualization, resources, supervision, and funding acquisition.
C Dieterich: conceptualization, resources, software, formal analysis, supervision, methodology, and writing—review and editing.
M Völkers: conceptualization, resources, formal analysis, supervision, funding acquisition, investigation, visualization, methodology, and writing—original draft, project administration, review, and editing.

## Conflict of Interest Statement

The authors declare that they have no conflict of interest.

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
