## [Reviewer comments · Life Science Alliance]

Life Science Alliance

m6A-mRNA methylation regulates cardiac gene expression and cellular growth

Vivien Kmietczyk, Eva Riechert, Laura Kalinski, Etienne Boileau, Ellen Malovrh, Brandon Malone, Agnieszka Gorska, Christoph Hofmann, Eshita Varma, Lonny Jürgensen, Verena Kamuf-Schenk, Janine Altmüller, Rewati Tappu, Martin Busch, Patrick Most, Hugo Katus, Christoph Dieterich, and Mirko Völkers

DOI: <https://doi.org/10.26508/lsa.201800233>

Corresponding author(s): Mirko Völkers, Heidelberg University Hospital

Review Timeline:

Submission Date:	2018-11-07
Editorial Decision:	2018-12-06
Revision Received:	2019-02-08
Editorial Decision:	2019-02-26
Revision Received:	2019-03-22
Accepted:	2019-03-22

Scientific Editor: Andrea Leibfried

Transaction Report:

December 6, 2018

Re: Life Science Alliance manuscript #LSA-2018-00233-T

Dr. Mirko Völkers
Heidelberg University Hospital
Internal Medicine, Cardiology
Im Neuenheimer Feld 410
Heidelberg, --- Select One --- 69120
Germany

Dear Dr. Völkers,

Thank you for submitting your manuscript entitled "m6A-mRNA methylation regulates cardiac gene expression and cellular growth" to Life Science Alliance. The manuscript was assessed by expert reviewers, whose comments are appended to this letter.

As you will see, the reviewers appreciate your data. However, they also note inconsistencies and would expect further support for your conclusions as well as some clarifications.

We would thus like to invite you to submit a revised version, addressing the reviewers' criticisms. Importantly, all reviewers note an inconsistency in the data overexpressing the enzyme-dead methyltransferase, and this issue needs to get resolved. Furthermore, the m6A measurements need to get clarified/rectified and re-evaluated (reviewer#1), and it would be good to provide a METTL3 expression analysis in human DCM tissues (reviewer #2). Reviewer #2 also thinks that the number of replicates is too low, but in case it is too difficult to obtain further human tissue samples for analysis, we'd advise you to follow reviewer #3's suggestion to represent the results as dot blots. Reviewer #3 points out that the validation of the meaningfulness of your results as exemplified on Arhgef3 and Myl2 needs to be more robust, and we agree with this view. This reviewer provides constructive input on how to do so. Finally, please discuss your work in light of the recently published competing work.

Thank you for this interesting contribution to Life Science Alliance. We are looking forward to receiving your revised manuscript.

Sincerely,

-- High-resolution figure, supplementary figure and video files uploaded as individual files: See our detailed guidelines for preparing your production-ready images, <http://life-science-alliance.org/authorguide>

B. MANUSCRIPT ORGANIZATION AND FORMATTING:

Full guidelines are available on our Instructions for Authors page, <http://life-science-alliance.org/authorguide>

*****IMPORTANT:** It is Life Science Alliance policy that if requested, original data images must be made available. Failure to provide original images upon request will result in unavoidable delays in publication. Please ensure that you have access to all original microscopy and blot data images

before submitting your revision.***

Reviewer #1 (Comments to the Authors (Required)):

In their manuscript, Kmietczyk et al. examine how m6A mRNA methylation affects gene expression in normal and diseased cardiac cells. They discover that changes to m6A methylation are correlated with cardiomyopathy. They demonstrate that changes to Mettl3 expression can affect cardiomyocyte growth and that these changes can be recapitulated in experiments that induce cardiac hypertrophy in mice. Finally, the authors perform analyses of sequencing data to examine how m6A methylation affects gene expression in cardiac cells.

Overall, the study provides some interesting insights into how m6A methylation regulates cell growth and gene expression in cardiomyopathy. While there are some discrepancies in the data (e.g. human DCM is associated with increased m6A methylation while induction of hypertrophy in mice results in decreased m6A methylation), the authors exercise appropriate caution in interpreting their results. While the results are somewhat preliminary and do not offer mechanistic insight into the m6A-mediated changes to cardiomyocyte function they observe, the results are interesting and well substantiated enough to merit publication. The sequencing data generated by the study could provide a nice resource to researchers interested in the role of m6A in cardiac cell biology. There are a few main points need to be clarified.

Major concerns:

1. In the experiments measuring m6A levels in RNA, the authors sometimes present m6A levels in total RNA (Fig 1A, Fig 2A and Ext Fig 2D) and sometimes present m6A levels in mRNA (Ext Fig 3A). What is the rationale for this choice? The authors must measure m6A on mRNA AND total RNA with more accurate approaches. Ideally LC-MS/MS. If only total RNA m6A level correlates with heart failure this most likely not goes through METTL3.
2. It is surprising that the authors can detect changes in m6A levels in total RNA upon overexpression or knockdown of Mettl3 (Fig 2A and Ext Fig 2D). Mettl3 affects only mRNA methylation, while >90% of total RNA is rRNA, and the 18S and 28S rRNAs are both m6A methylated. Furthermore, m6A is present in other abundant RNA species such as the U2, U4 and U6 snRNAs. In the analysis of human samples (Fig 1A), is it possible that the changes in m6A methylation are due to the altered methylation of species other than mRNAs? This data is not consistent with proposed pathway and need to be carefully re-evaluated.
3. The authors use antibody-based detection of m6A in a commercial ELISA kit. Because anti-m6A antibodies cross-react with N6-methyl deoxyadenosine, a DNA modification commonly found in bacterial DNA, it is important to ensure that cell line samples are free from common bacterial contaminants like mycoplasma and/or perform DNase digestion during total RNA purification. Were these steps taken?

Minor concerns:

4. References are often wrong. The FTO discovery was made in 2011, followed by m6A-seq in 2012. The reversible RNA methylation was proposed in 2010. For FTO activity see recent Mol. Cell paper: [https://www.cell.com/molecular-cell/pdfExtended/S1097-2765\(18\)30645-2](https://www.cell.com/molecular-cell/pdfExtended/S1097-2765(18)30645-2); discussions on page 4 last paragraph is inappropriate without citing recent literature.

5. In Fig 2A, overexpression of mutant Mettl3 increases m6A levels, which is not expected. The authors claim that the mutant is enzymatically dead, but they do not specify the mutation(s) introduced into Mettl3 to inactivate the protein. Can the authors provide information about the Mettl3 mutant the used? Is it possible that the Mettl3 mutant exhibits partial activity?
6. Line 136. The authors claim that overexpression of the inactive Mettl3 mutant did not affect cell size, yet in Fig 2B, they show a statistically significant difference between AdCo + PE and AdMut + PE ($p < 0.005$).
7. Line 146. I think the reference should be to Ext. Fig 2H, not 2F.
8. Line 179. I think the reference should be to Ext. Fig 3A, not 2A.
9. Ext. Fig 3D. The left panel is missing the label for the x-axis.
10. Line 231-233: "We speculate that Mettl3 affects translational efficiency by methylation mRNAs encoding for proteins involved in transcriptional regulation which could fine-tune the response to cellular stress." Other explanations are also possible. For example, various m6A reader proteins are known to regulate the translation of methylated transcripts, including during stress. These readers often target specific transcripts and their activity can vary between different cell types. The altered translation efficiency of m6A methylated transcripts in TAC-operated mice could potentially be explained by stress-induced changes to the expression of m6A readers like Ythdf1.

Reviewer #2 (Comments to the Authors (Required)):

The manuscript by Kmietczyk et al describes a novel role for cardiac m6A- and METTL3-mediated post-transcriptional regulation. The authors report increased m6A RNA methylation and METTL3 expression in human dilated cardiomyopathy (DCM). Using MeRIP-seq, the authors further identify that majority of m6A-methylated mRNAs belong to transcriptional pathways and unmethylated RNAs belong to translational pathways in DCM. Further, they show that METTL3 knockdown increases cardiomyocyte hypertrophy (in vitro) whereas METTL3 overexpression decreases myocyte hypertrophy both in vivo and in vitro. Finally, they demonstrate using Ribo-seq and RNA-seq approaches that changes in m6A methylome mediated by METTL3 regulates translational control in hypertrophic (TAC) hearts. Overall, the study addresses a novel and interesting regulatory mechanism working at the post-transcriptional level in the heart, however, several data presented are preliminary to derive strong conclusions for publication in the journal.

Comments:

- Western blots assessing METTL3 expression in human DCM tissues are required. This is especially important given the main focus of the manuscript based on METTL3 and only small sample sizes are included for mRNA quantification in DCM tissues. Number of samples are not shown in figure legend for mRNA quantification, however I assume $N=2-3$ based on Fig. 1A, which is too little to conclude on METTL3 expression. This would be needed to obtain statistical significance for Mettl3 expression especially for journal publication.
- In addition, many of the figures (eg. Fig 2A, B) have sample sizes too low and in certain cases only $N=2$, which seems too preliminary for journal publication. The authors should increase samples size numbers to assess statistical significance in many instances.

- The authors report that loss of METTL3 increases cardiomyocyte hypertrophy in vitro while METTL3 overexpression decreases hypertrophy both in vitro and in vivo. How does this observation compare and contrast to a recent report (Lisa E Dorn et al., Circulation; 28 Nov 2018) that demonstrates a role for METTL3 in cardiac hypertrophy? The authors can include a discussion point on this.
- Overexpression of mutant METTL3 (enzymatically inactive) results in significant increase in m6A although slightly lower than active METTL3 overexpression. How do the authors explain this increase with mutant METTL3? Is the mutant METTL3 a completely inactive mutant or still a leakage of methyltransferase activity present in these constructs? Can the authors confirm this experimentally? Alternatively discuss.
- The authors study METTL3 overexpression mouse models of TAC. Could they provide the level of m6A RNA methylation in mouse models overexpressing METTL3 to show if METTL3 was sufficient to increase m6A levels in mouse hearts?
- Authors argue that highly translated transcripts in METTL3 overexpressing cardiac myocytes were highly translated in response to TAC surgery and vice versa. Does this mean increased RNA methylation by METTL3 serves to increase mRNA translation to protein? How does this fit with TAC model when there is a global reduction in total m6A level at two days post TAC surgery? Is Myl2 highly methylated in TAC thus more Myl2 protein? Can authors also provide western blots showing Myl2 protein increase in Mettl3 overexpressing myocytes as well as in TAC hearts? How does this compare when in DCM, there is increased METTL3 expression, however only low m6A containing RNAs belong to translational pathway?
- What is the expression level of METTL3 (RNA/protein) in TAC hearts as compared to sham hearts? Do they observe changes to METTL3 expression in the setting of cardiac hypertrophy in mouse?
- Please correct dilative to dilated in page 3.
- There seems a color mismatch for Nde1 in Ext. Fig. 1. Please correct.
- Ext fig 1D is not mentioned in the manuscript
- On page 6, line 179, Extended Fig. 2A was mislabeled instead it should be 3A.
- Can the authors explain the methods of bioinformatics analysis of m6A-seq in more detail?

Reviewer #3 (Comments to the Authors (Required)):

m6A-mRNA methylation regulates cardiac gene expression and cellular growth

Overall, this is an interesting and timely manuscript. The field of epitranscriptomics is growing quickly, and little is known about the role of m6A in the heart. The Authors find that RNA m6A levels

are increased in failing human hearts, and that the distribution of m6A different in failing and non-failing samples. They go on to use NRVMs, HL1, and mice to focus on how METTL3 expression regulates m6A levels on cardiac RNA, and how this relates to relevant cardiac phenotypes. Their data show that METTL3 knockdown (loss of m6A) leads to greater cardiac hypertrophy, while METTL3 overexpression (increase of m6A) reduces pathologic cardiomyocyte hypertrophy and fibrosis. Interestingly, cardiac m6A levels are reduced in the acute phase of TAC stress (2d), which is the opposite of what was seen in failing human hearts.

Major Critique

In an effort to assess how METTL3 might be regulating these phenotypes, the authors use Ribo-seq in both the animal model (TAC) and cells (HL1). The results here are somewhat confusing and need to be clarified. I think the authors are trying to show that m6A-enriched mRNA transcripts have a higher ribosome occupancy, suggesting higher rates of translation, while mRNA levels are unchanged (Fig 3B and 3E). However, polysome profiling was no different between METTL3 overexpression and control (Fig 3F). Select transcripts did have METTL3-dependent (and presumably m6A-dependent) association with polysomes (Fig 3G). At this point in the manuscript, it is not even clear what material is being assayed (HL1 cells vs mouse hearts; most likely HL1 cells). Finally, the Authors show that there are differences in RNA stability in the control and METTL3-overexpression states, for the same transcripts that show differential polysome loading. Since the Authors have identified *Arhgef3* and *Myl2* for further study as exemplary genes being regulated by m6A, I suggest they show the following: quantitative Ribo-seq, RNA-seq, and m6A-RIP data for these transcripts, and immunoblotting to demonstrate their protein expression under the control and experimental condition. Furthermore, the Authors should discuss how the results of their data explain the expression of these two genes, and how this relates to m6A-dependent regulation of cardiomyocyte hypertrophy or cardiac fibrosis. Alternative genes could be chosen for this analysis, but the important point is to provide deeper evidence that m6A regulates expression of at least one relevant gene in the heart. For example: *Myl2* mRNA seems to be stabilized by m6A, and there is more *Myl2* mRNA associated with polysomes. Is there actually more *Myl2* protein? If so, is it because there is more RNA, or more efficient translation, or both? Then a discussion of how increased *Myl2* expression fits into the overall picture of m6A and the cardiomyocyte hypertrophic response.

Other Issues

Beyond this major issue, there are a surprisingly large number of other issues that suggest a lack of attention to detail in reporting the findings. The following issues should be also addressed:

1. Line 73: The authors state that it is "conceivable that m6A regulates translation". This is well-established in many publications, including Ref #11. Please revise the manuscript text to reflect the current state of knowledge and cite additional references.
2. Ref #17 and #18 are same. Please consolidate these.
3. ExtFig 1B, IP fraction: Both bars are red; I assume this is an error. Please correct.
4. Figure 1A and 1B: Since the sample size is very small (n= 2 control, and n=3 DCM), please use dot plots to show each data point. Also, error bars must be defined for ALL figures (SEM, SD, ?). "*" is used to denote significance, but this is not described in the Figure Legend (?p<0.05). Please ensure that ALL figures denote the meaning of such symbols. Finally, since these are human tissue samples, please include how they were obtained, and cite IRB approval for this.

5. Line 100: Authors state that "thousands of genes were significantly enriched in the IP fraction from failing myocardium". Does this refer to the ~2400 genes shown in Table 1 and Fig 1C? In the Methods, the Authors define enrichment as $\log FC > 0$ (and $FDR < 0.05$). Applying these parameters to Table 1 DCM samples identifies 2480 genes as "enriched" in the m6A IP sample compared to the control Ab. Notably, this number differs somewhat from the number of "m6A enriched genes in DCM" shown in Fig 1C ($1518 + 877 = 2395$). A similar difference is noted for the Ctrl samples (1216 "enriched" genes applying the stated statistical criteria to Table 1, and $304 + 877 = 1181$ genes shown in Fig 1C). Please clarify how exactly how many genes were identified as "m6A enriched" in the human samples, and reconcile Table 1 with Fig 1C.
6. Tables 1-4 are included and referenced but there are no Legends to describe them. Please provide Legends and label the Table headers to indicate which samples are m6A Ab IP vs the ctrl IP.
7. Fig 1F: Color scheme used here is opposite of that in Fig 1A, 1B, and 1C; this is very confusing. Please use a consistent scheme throughout (e.g., blue = ctrl, red = DCM).
8. Ext.Fig 1D is not referenced in the text. Please delete this or reference it in the text.
9. ExtFig 2: Please show scale bars for photomicrographs. ExtFig 2C: Two panels are shown, but not labeled. Which one is METTL3 vs inactive mutant?
10. Fig 2A: Please explain why the catalytically-dead METTL3 overexpression increases m6A levels. The text of line 154 is not a true statement given the data presented (dead METTL3 did change m6A levels, but it did not alter growth response).
11. Fig 2B: Why does METTL3 overexpression not reduce cell size, as it does in the heart (Fig 1F)? In Fig 2B, the bar graphs for AdCo, AdMe, and AdMut look identical, including their error bars. The Y-axis is labeled as "relative CSA n-fold change vs AdCo". Please provide the raw data for review.
12. Fig 2C-I: What was the AAV "control" used? If it was not the catalytically dead METTL3, why not (given that this was used in Fig 2A and B).
13. Fig 2E: There are four panels; are the right-hand panels +TAC?
14. ExtFig 2H is not mentioned in the text. It should be referenced on line 144.
15. Line 167: "Surprisingly, 50% less overall m6A levels were measured...." The Figure shown does not support this statement. Please state the measured value.
16. Ext Fig 3B and Table 2: Again, there is a discrepancy between the Figure and the Table. Based on the criteria stated in the methods ($\log FC > 0$ and $FDR < 0.05$), the Table indicates 1567 "m6A enriched" genes in the sham set, but the Figure and Text refer to 1543 genes. Table shows 330 "m6A enriched" genes in the TAC samples, but the Text and Figure note $206 + 155 = 361$. Please explain/clarify.
17. Fig 3B: Please clarify how "Ribo-seq \log_2 -Fold change" is calculated (normalized Ribo-seq reads per transcript, sham/TAC?). Please also clarify how why the X-axis is showing both Sham and TAC treatments, while the Legend states that each box plot represents Sham/TAC fold-change. Perhaps the left box plot is for "non-m6A enriched genes" and the right is for "m6A enriched genes"?

18. Ext Fig 3D: Left panel is not labeled; presumably Ribo-seq data.

19. Line 187: Here, the Authors introduce a new model: HL1 cells. This needs to be clearly stated in the text of the manuscript.

20. Line 196: Table 1 is referenced here; I presume this should actually be Table 3.

Other than the criticisms listed here, the Authors should be commended for a thorough and detailed Methods section.

Dr. Andrea Leibfried
Executive Editor
Life Science Alliance

Dear Dr. Leibfried

Thank you for inviting us to revise our submission "m⁶A-mRNA methylation regulates cardiac gene expression and cellular growth". We thank the reviewers for their careful analysis of our manuscript. We appreciate the reviewers' recognition that we set up an interesting study how m⁶A-methylation regulates cell growth in cardiac myocytes. We believe we have addressed all reviewer concerns and have significantly improved the quality of our submission. We hope you will now find our work acceptable for publication in Life Science Alliance. Our specific responses and references to changes in the revised manuscript are delineated as follows:

Response to Reviewers:

Reviewer #1: *In their manuscript, Kmietczyk et al. examine how m⁶A mRNA methylation affects gene expression in normal and diseased cardiac cells. They discover that changes to m⁶A methylation are correlated with cardiomyopathy. They demonstrate that changes to Mettl3 expression can affect cardiomyocyte growth and that these changes can be recapitulated in experiments that induce cardiac hypertrophy in mice. Finally, the authors perform analyses of sequencing data to examine how m⁶A methylation affects gene expression in cardiac cells. Overall, the study provides some interesting insights into how m⁶A methylation regulates cell growth and gene expression in cardiomyopathy. While there are some discrepancies in the data (e.g. human DCM is associated with increased m⁶A methylation while induction of hypertrophy in mice results in decreased m⁶A methylation), the authors are exercise appropriate caution in interpreting their results. While the results are somewhat preliminary and do not offer mechanistic insight into the m⁶A-mediated changes to cardiomyocyte function they observe, the results are interesting and well substantiated enough to merit publication. The sequencing data generated by the study could provide a nice resource to researchers interested in the role of m⁶A in cardiac cell biology. There are a few main points need to be clarified.*

We appreciate the reviewer's assessment and have addressed the comments as follows:

Major concerns:

1. *In the experiments measuring m⁶A levels in RNA, the authors sometimes present m⁶A levels in total RNA (Fig 1A, Fig 2A and Ext Fig 2D) and sometimes present m⁶A levels in mRNA (Ext Fig 3A). What is the rationale for this choice? The authors must measure m⁶A on mRNA AND total RNA with more accurate approaches. Ideally LC-MS/MS. If only total RNA m⁶A level correlates with heart failure this most likely not goes through METTL3.*

We totally agree with the reviewer that we must measure m⁶A in mRNA in human heart failure. We initially performed measurement of m⁶A-levels on total RNA in human heart failure samples because of the limited amount of tissue and RNA isolated from the diseased human tissue. We accidentally mislabeled graphs about m⁶A levels from isolated cells with overexpression and knock-down of Mettl3 (Fig 2A, Ext Fig 2D) and apologize for any confusions. Measurement of m⁶A-levels from isolated cells have been performed on mRNA (not total RNA).

To address this important concern about the human data, additional experiments have been incorporated into the revised manuscript. Specifically, new measurements of m⁶A levels have been performed on mRNA from human heart failure samples and following the suggestion from Reviewer #2 also from mice with overexpression of Mettl3. In addition, we also increased the n-numbers of m⁶A measurements from mRNAs after overexpression of mutant Mettl3- see also our response to your point 5.

m⁶A level in mRNA increase in human heart failure and are higher in hearts with increased expression of Mettl3. Similar reports have been shown in a published reports using the commercially available ELISA (Mathiyalagan et al, 2018; Dorn et al, 2018). We also agree that measurement with

LC-MS would be the most accurate approach, however because of the limited resources of human (diseased) tissue we were not able to perform additional LC-MS based quantification of m⁶A levels. The revised information is now shown in new the Figure panels and mentioned in the revised manuscript as follows:

Page 4, line 103

In line with a recently published report (Mathiyalagan *et al*, 2018), we found increased m⁶A levels in mRNAs (Fig 1A) isolated from human failing myocardium

Page 14, line 471:

Total RNA was isolated from cultured cell lysates by using the Quick-RNA™ MiniPrep Kit from Zymo Research including an on gDNA removal step and an additional In-column DNase I digest step. mRNA was isolated by polyA enrichment using oligo(dt) magnetic beads (NEB) according to the manufacturer's instructions.

2. *It is surprising that the authors can detect changes in m6A levels in total RNA upon overexpression or knockdown of Mettl3 (Fig 2A and Ext Fig 2D). Mettl3 affects only mRNA methylation, while >90% of total RNA is rRNA, and the 18S and 28S rRNAs are both m6A methylated. Furthermore, m6A is present in other abundant RNA species such as the U2, U4 and U6 snRNAs. In the analysis of human samples (Fig 1A), is it possible that the changes in m6A methylation are due to the altered methylation of species other than mRNAs? This data is not consistent with proposed pathway and need to be carefully re-evaluated.*

We again apologize for the confusion and mis-labeling of the Fig.2A and Ext Fig 2D. We performed measurement of m⁶A-levels from isolated cardiac myocytes from purified **mRNA**, not total RNA (see also our response to your point #1). Since we have repeated the m⁶A measurement of human heart failure samples, we have also included additional measurement of m⁶A levels from isolated cardiac myocytes and per request of reviewer #2 new measurements from hearts with Mettl3 overexpression. These data now clearly show that m⁶A level in mRNA is increased after Mettl3 overexpression and vice versa in cardiac myocytes and are in line with recent published reports about FTO in cardiac myocytes and Mettl3. Still, we cannot rule out from our previous dataset in human disease tissue (where we measured total RNA) that methylation of other RNA species is changed in diseased human hearts.

3. *The authors use antibody-based detection of m6A in a commercial ELISA kit. Because anti-m6A antibodies cross-react with N6-methyl deoxyadenosine, a DNA modification commonly found in bacterial DNA, it is important to ensure that cell line samples are free from common bacterial contaminants like mycoplasma and/or perform DNase digestion during total RNA purification. Were these steps taken?*

We appreciate the opportunity to address this excellent question and again agree that it is important to perform DNA digestion. Indeed, Total RNA was isolated from cultured cell lysates by using the Quick-RNA™ MiniPrep Kit from Zymo Research including an on gDNA removal step and an additional In-column DNase I digest step. Moreover, we routinely test our cell lines for mycoplasma contamination. The revised information is now incorporated into the Methods section as follows.

Page 14, line 468:

Total RNA was isolated from cultured cell lysates by using the Quick-RNA™ MiniPrep Kit from Zymo Research including an on gDNA removal step and an additional In-column DNase I digest step. mRNA was isolated by polyA enrichment using oligo(dt) magnetic beads (NEB) according to the manufacturer's instructions.

Minor concerns:

4. *References are often wrong. The FTO discovery was made in 2011, followed by m6A-seq in 2012. The reversible RNA methylation was proposed in 2010. For FTO activity see recent*

Mol. Cell paper: [https://www.cell.com/molecular-cell/pdfExtended/S1097-2765\(18\)30645-2](https://www.cell.com/molecular-cell/pdfExtended/S1097-2765(18)30645-2); discussions on page 4 last paragraph is inappropriate without citing recent literature.

We apologize for the mistakes. As a consequence, we have revised the manuscript to include comprehensive information about m⁶A, FTO and m⁶A-seq and added relevant citations in the discussion.

Page 2, line 58

Reversible mRNA modifications have been proposed in 2010 by the He laboratory (He, 2010) and the discovery of fat mass and obesity-associated (Fto) and AlkB Homolog 5 RNA Demethylase (Alkbh5) proteins as m⁶A demethylases in 2011 has finally shown the dynamic, reversible, and adjustable nature of m⁶A RNA modifications (Jia *et al*, 2011)

Page 5, line 141

Fto binds additional RNA species including snRNA and tRNA (Wei *et al*, 2018) and recent reports showed that Fto also demethylates N6,2'-O-dimethyladenosine (m⁶Am) as well as snRNAs or tRNAs in addition to m⁶A (Mauer *et al*, 2017; Wei *et al*, 2018). Thus, we focused on studying Mettl3 in further experiments.

5. *In Fig 2A, overexpression of mutant Mettl3 increases m6A levels, which is not expected. The authors claim that the mutant is enzymatically dead, but they do not specify the mutation(s) introduced into Mettl3 to inactivate the protein. Can the authors provide information about the Mettl3 mutant the used? Is it possible that the Mettl3 mutant exhibits partial activity?*

The reviewer raises another good point. Similar points have been raised also by Reviewer #2 and #3. Two mutations were induced within the catalytic region (CMII) of Mettl3 in order to change amino acid structure. Mutation 1 changed asparagine (AA 395) to Alanine, mutation 2 changes Tryptophan (AA 398) to Alanine, resulting in a mutant Mettl3 lacking methyltransferase activity (aa395-398, DPPW → APPA). This has been shown in previous publications to completely abolish Mettl3 enzymatic activity (Vu *et al*, 2017; Alarcón *et al*, 2015).

To address this important concern, we have included additional measurement of m⁶A levels from isolated cardiac myocytes after Mettl3 wildtype and mutant overexpression. These data are now shown in the revised Figure 2A and now show only a small but not significant increase in m⁶A-levels in myocytes overexpressing mutant Mettl3. We cannot fully rule out that in our cellular system the Mettl3 mutant exhibits partial activity, but these data now suggest that our mutant Mettl3 construct indeed blocks the enzymatic activity in cardiac myocytes.

The revised information is now shown in new Figures 2A and mentioned in the revised manuscript as follows:

Page 5, line 144:

Overexpression of the enzymatically active Mettl3 significantly increased m⁶A levels, whereas an enzymatically dead mutant of Mettl3 (Vu *et al*, 2017; Alarcón *et al*, 2015) did not increase m⁶A levels compared to the control group (Fig 2A).

Page 13, line 451:

Generation of the mutant Mettl3 (Mettl3 Mut) construct.

2 point mutations were introduced within the catalytic site of Mettl3 (Vu *et al*, 2017) in two mutagenesis steps by using the following PCR protocol with pAd_Mettl3 as a parent plasmid and specifically designed primers containing the desired mutations using the Agilent QuikChange II Site-Directed Mutagenesis Kit per manufacturers instruction. The obtained plasmid was transformed into NEB alpha-5 E. Coli following the High Efficiency Transformation Protocol from New England Biolabs. Validation of the created plasmid was performed by sequencing and the mutant construct was cloned into a pAd vector using the LR Recombination protocol by Thermo Fisher Scientific.

6. Line 136. The authors claim that overexpression of the inactive *Mettl3* mutant did not affect cell size, yet in Fig 2B, they show a statistically significant difference between AdCo + PE and AdMut + PE ($p < 0.005$).

The reviewer is correct. As the reviewer points out, overexpression of mutant *Mettl3* indeed significantly increased cell size compared to the control Adenovirus in response to neurohumoral stimulation with Phenylephrine. The reason for this is currently unknown. Altered methylation pattern of transcripts (by acting as a dominant negative mutant) could potentially explain this observation as well as altered binding to alternative binding partners to the mutant *Mettl3*. We revised the manuscript and this point was added to the results and discussion.

Page 5, line 149:

In contrast, overexpression of enzymatically inactive *Mettl3* mutant did not block cell growth in response to PE, but rather augmented cellular size compared to control cells.

7. Line 146. I think the reference should be to Ext. Fig 2H, not 2F.
8. Line 179. I think the reference should be to Ext. Fig 3A, not 2A.
9. Ext. Fig 3D. The left panel is missing the label for the x-axis.

The reviewer is correct. We have extensively reviewed the manuscript and made sure that the manuscript does not contain the mentioned errors.

10. Line 231-233: "We speculate that *Mettl3* affects translational efficiency by methylation mRNAs encoding for proteins involved in transcriptional regulation which could fine-tune the response to cellular stress."

Other explanations are also possible. For example, various m⁶A reader proteins are known to regulate the translation of methylated transcripts, including during stress. These readers often target specific transcripts and their activity can vary between different cell types. The altered translation efficiency of m⁶A methylated transcripts in TAC-operated mice could potentially be explained by stress-induced changes to the expression of m⁶A readers like *Ythdf1*.

We concur with the reviewer that other mechanisms could be possible and explain the observed phenotypes. Clearly, m⁶A reader proteins are known regulators of translational efficiency of methylated transcripts. This point was added to the discussion.

Page 9, line 280

Alternatively, since m⁶A-reader like *Ythdf1-3* (Wang *et al*, 2015; Shi *et al*, 2018, 2017) can regulate translational efficiency of methylated transcripts, especially in cellular stress conditions, changes in expression or activity of those readers might alter translational efficiency of methylated transcripts

We thank the reviewer for the erudite commentary and hope these revisions serve to appropriately address the provided critique.

Reviewer #2: The manuscript by Kmiotczyk *et al* describes a novel role for cardiac m⁶A- and METTL3-mediated post-transcriptional regulation. The authors report increased m⁶A RNA methylation and METTL3 expression in human dilated cardiomyopathy (DCM). Using MeRIP-seq, the authors further identify that majority of m⁶A-methylated mRNAs belong to transcriptional pathways and unmethylated RNAs belong to translational pathways in DCM. Further, they show that METTL3 knockdown increases cardiomyocyte hypertrophy (in vitro) whereas METTL3 overexpression decreases myocyte hypertrophy both in vivo and in vitro. Finally, they demonstrate using Ribo-seq and RNA-seq approaches that changes in m⁶A methylome mediated by METTL3 regulates translational control in hypertrophic (TAC) hearts. Overall, the study addresses a novel and interesting regulatory mechanism working at the post-transcriptional level in the heart, however, several data presented are preliminary to derive strong conclusions for publication in the journal.

We thank the reviewer for these supportive remarks and have addressed comments as follows:

Comments:

1. *Western blots assessing METTL3 expression in human DCM tissues are required. This is especially important given the main focus of the manuscript based on METTL3 and only small sample sizes are included for mRNA quantification in DCM tissues. Number of samples are not shown in figure legend for mRNA quantification, however I assume N=2-3 based on Fig. 1A, which is too little to conclude on METTL3 expression. This would be needed to obtain statistical significance for Mettl3 expression especially for journal publication.*

This is an important point. To address this, we performed as requested Immuno-blots for Mettl3 expression in human DCM tissue. Additionally, we analyzed mRNA transcript levels in human biopsies from recently published DCM cohort (n=33) by RNA-seq compared to healthy controls (n=24)(Meder *et al*, 2017). These data now show no significant changes in RNA-seq read counts in DCM hearts compared to healthy control hearts. However, we do see a small, albeit not significant increase in Mettl3 protein levels in the Immunoblots from DCM hearts, which could suggest additional post-transcriptional regulation of Mettl3 expression including regulation at the level of transcript stability or degradation. These data are now shown in Ext. Fig 1C.

We revised the manuscript and this point was added to the results and discussion.

Page 4, line 105

We analyzed expression of m⁶A-writers and erasers in human heart biopsies from a published DCM patient cohort (n=33) compared to healthy controls (n=24) by RNA-seq (Meder *et al*, 2017). These data did not show any significant changes of Mettl3 or FTO on the transcript level (Fig 1B). A small, but not significant increase in protein levels of Mettl3 could be observed in DCM hearts compared to control samples (Ext.Fig 1C).

2. *In addition, many of the figures (eg. Fig 2A, B) have sample sizes too low and in certain cases only N=2, which seems too preliminary for journal publication. The authors should increase samples size numbers to assess statistical significance in many instances.*

Following the reviewer suggestion, we performed the additional experiments including m⁶A measurements, cell size experiments and PCR- analysis. These data are now incorporated into the revised manuscript and mentioned in the figure legends.

3. *The authors report that loss of METTL3 increases cardiomyocyte hypertrophy in vitro while METTL3 overexpression decreases hypertrophy both in vitro and in vivo. How does this observation compare and contrast to a recent report (Lisa E Dorn et al., Circulation; 28 Nov 2018) that demonstrates a role for METTL3 in cardiac hypertrophy? The authors can include a discussion point on this.*

The reviewer raises another good point. During review of our manuscript, a manuscript was published in *Circulation*. In this manuscript the authors also observed increased methylation levels in myocytes after overexpression with Mettl3, which was associated with **increased** cell growth. *In vivo*, transgenic mice do show increased heart weight, but response to TAC surgery was unchanged.

Differences in the study design could explain some of the contrasting data (blocked hypertrophy after overexpression in our *in vivo* model). Dorn et al. used a transgenic model in the FVB background, where we used a C57Bl/N background for the *in vivo* studies. Moreover, we used an AAV based approach to overexpress Mettl3 which could result in differences in expression levels compared to a transgenic approach driven by the aMHC-promoter in the Dorn et al. study. In line, increased hypertrophy in the Mettl3 transgene was only observed in the higher expressing transgenic line (mouse line 2 in Fig. 2F in the Lisa E Dorn et al., *Circulation*; 28 Nov 2018 manuscript).

We revised the manuscript and this point was added to the discussion.

Page 5, line 170

Very recently an elegant report analyzed the role of Mettl3 during hypertrophic cardiac growth by using cardiac-restricted gain- and loss-of-function mouse models (Dorn et al, 2018). This study showed- in contrast to our study- that increased Mettl3 expression caused (compensated) hypertrophy in vivo, whereas response to TAC was unchanged. Differences in the study design could explain some of the contrasting data. Dorn et al. used a transgenic model in the FVB background, where we used a C57Bl6/N background for the in vivo studies. Moreover, we used an AAV based approach to overexpress Mettl3, which resulted in lower overexpression levels compared to a transgenic approach driven by the aMHC-promoter in the Dorn et al. study. However, both studies point towards an important role of epitranscriptomic control on cell growth. More studies are clearly needed to fully understand this novel stress-response mechanism in the heart for maintaining normal cardiac function.

4. *Overexpression of mutant METTL3 (enzymatically inactive) results in significant increase in m6A although slightly lower than active METTL3 overexpression. How do the authors explain this increase with mutant METTL3? Is the mutant METTL3 a completely inactive mutant or still a leakage of methyltransferase activity present in these constructs? Can the authors confirm this experimentally? Alternatively discuss.*

An excellent point that has been also raised by by Reviewer #1 and #3. Two mutations were induced within the catalytic region (CMII) of Mettl3 in order to change amino acid structure. Mutation 1 changed asparagine (AA 395) to Alanine, mutation 2 changes Tryptophan (AA 398) to Alanine, resulting in a mutant Mettl3 lacking methyltransferase activity (aa395-398, DPPW → APPA). This has been shown in previous publications to completely abolish Mettl3 enzymatic activity (Vu et al, 2017; Alarcón et al, 2015).

To address this important concern, we have included additional measurement of m⁶A levels from isolated cardiac myocytes after Mettl3l wildtype and mutant overexpression- see also our response to your point #2. These data are now shown in the revised Figure 2A and now show only a small but not significant increase in m⁶A-levels in myocytes overexpressing mutant Mettl3. We cannot fully rule out that in our cellular system the Mettl3 mutant exhibits partial activity, but these data now suggest that our mutant Mettl3 construct indeed blocks enzymatic activity in cardiac myocytes. To confirm this experimentally, *in vitro* methylation assay using recombinant Wild-type Mettl3 and Mutant would be necessary, which is a challenging and complicated experiment. We believe that this would be out of the scope of the current manuscript and we added the new data in to revised manuscript together with information about the mutated Mettl3 construct.

The revised information is now shown in new Figures 2 and mentioned in the revised manuscript as follows:

Page 5, line 144:

Overexpression of the enzymatically active Mettl3 significantly increased m⁶A levels, whereas an enzymatically dead mutant of Mettl3 (Vu et al, 2017; Alarcón et al, 2015) did not increase m⁶A levels compared to the control group (Fig 2A).

Page 13, line 451:

Generation of the mutant Mettl3 (Mettl3 Mut) construct.

2 point mutations were introduced within the catalytic site of Mettl3 (Vu et al, 2017) in two mutagenesis steps by using the following PCR protocol with pAd_Mettl3 as a parent plasmid and specifically designed primers containing the desired mutations using the Agilent QuikChange II Site-Directed Mutagenesis Kit per manufacturers instruction. The obtained plasmid was transformed into NEB alpha-5 E. Coli following the High Efficiency Transformation Protocol from New England Biolabs. Validation of the created plasmid was performed by sequencing and the mutant construct was cloned into a pAd vector using the LR Recombination protocol by Thermo Fisher Scientific.

5. *The authors study METTL3 overexpression mouse models of TAC. Could they provide the level of m⁶A RNA methylation in mouse models overexpressing METTL3 to show if METTL3 was sufficient to increase m⁶A levels in mouse hearts?*

We concur with the reviewer and provide new evidence to support the postulate that Mettl3 increases m⁶A level in mouse hearts. m⁶A levels in mRNAs are increased in AAV-Mettl3 mice hearts measured by additional m⁶A-Elisas. These data are now included in the manuscript and mentioned in the text as delineated below.

Page 5, line 157:

Overexpression increased m⁶A-levels in mRNAs isolated from mice hearts (Ext.Fig 2H)

6. *Authors argue that highly translated transcripts in METTL3 overexpressing cardiac myocytes were highly translated in response to TAC surgery and vice versa. Does this mean increased RNA methylation by METTL3 serves to increase mRNA translation to protein? How does this fit with TAC model when there is a global reduction in total m⁶A level at two days post TAC surgery? Is Myl2 highly methylated in TAC thus more Myl2 protein? Can authors also provide western blots showing Myl2 protein increase in Mettl3 overexpressing myocytes as well as in TAC hearts? How does this compare when in DCM, there is increased METTL3 expression, however only low m⁶A containing RNAs belong to translational pathway?*

This is an excellent question and we have performed new experiments to answer the reviewer questions (see also major comment from reviewer #3). Indeed, we do see a trend to increased Myl2 methylation 2 days post TAC, which was not significant in our data set (log₂FC=0.2 compared to total RNA, p= 0.1), whereas Myl2 transcript is depleted in our m⁶A-seq data in sham operated mice (log₂FC= - 0.23 compared to total RNA, p=3.62E-14).

This could indeed suggest that Myl2 transcripts are higher methylated early after TAC compared to Sham operated mice. In human heart failure samples, Myl2 transcripts are depleted after m⁶A-IP compared to the input total RNA. The detailed consequences of Myl2 transcript methylation are just unknown as how changing m⁶A levels are explained in a complex interplay of writers and eraser in failing myocardium. At the moment we cannot fully explain the decreased overall m⁶A -levels two days post TAC surgery. Whether this is due to altered Mettl3 or Fto enzymatic activity, needs to be investigated. We do observe a small decrease in Mettl3 protein expression 2 days post TAC (new Ext. Fig. 3C). In addition, subcellular localization of either Mettl3 or FTO could change early in response to TAC surgery. Finally, Methylation is an energy consuming process and it is unknown, how enzymatic activity of Mettl3 is regulated

Still, new experiments in isolated myocytes show now that Mettl3 overexpression increased Myl2 and Arhgef3 methylation (measured with RIP-PCR), which indeed resulted in increased protein levels of Myl2, and decreased protein levels of Arhgef3 whereas mRNA transcript levels remained unchanged. These data now suggest that Mettl3 dependent methylation of regulates protein translation independent of changes in the total mRNA levels of specific target mRNAs in cardiac myocytes.

In vivo, the situation seems to be more complex. Following your suggestion to assess Mettl3 expression in TAC hearts (see point #7) we measured RNA and protein levels of Mettl3 in TAC hearts. Interestingly, we do observe a small decrease in Mettl3 RNA and protein levels 2-day post TAC surgery which is associated with the decrease in RNA-methylation at this time point.

In contrast, a slight increase in Mettl3 expression is observed at 2 weeks post TAC surgery, and Myl2 protein expression is increased, but Mettl3 overexpression almost completely blocked induction of Myl2 expression. Increased Myl2 protein levels are also observed in DCM patients.

These data are now included in the manuscript and mentioned in the text as delineated below.

Page 7, line 234:

We validated Mettl3-dependent translation of two candidates (*Arhgef3* and *Myl2*) by qRT-PCR on polysomal fractions from control or Mettl3 overexpressing cardiomyocytes. Overall translation was unchanged in Mettl3 overexpressing myocytes as assessed by polysome profiles (Fig 4A). However, *Arhgef3* was significantly less while *Myl2* more translated in our HL-1 Ribo-seq data. In contrast, mRNA levels remain unchanged after Mettl3 overexpression (Fig. 4B). As predicted by our Ribo-seq data, *Arhgef3* transcript levels were decreased in polysomal fractions, whereas *Myl2* levels increased after Mettl3 overexpression (Fig 4C). Increased Mettl3 dependent methylation of *Arhgef3* and *Myl2* transcripts were validated by qRT-PCR after m⁶A-precipitated mRNA from Mettl3- or control-HL-1 cells compared to the input mRNA (Fig.4D).

Page 7, line 243:

Finally, Mettl3 overexpression caused decreased *Arhgef3* protein levels, whereas *Myl2* levels were increased in Mettl3 overexpressing myocytes (Fig. 4F-G). In contrast to our *in vitro* data, mice overexpressing Mettl3 in the heart showed decreased *Myl2* expression both during sham and TAC conditions. We speculate that longer Mettl3 overexpression results in different methylation patterns *in vivo* compared to our *in vitro* data. Moreover, different expression levels of m⁶A reader proteins could certainly affect translation of methylated transcripts after long-term overexpression of Mettl3. Ultimately, more studies are needed to fully understand how m⁶A methylation regulates translation in myocytes.

7. *What is the expression level of METTL3 (RNA/protein) in TAC hearts as compared to sham hearts? Do they observe changes to METTL3 expression in the setting of cardiac hypertrophy in mouse?*

An excellent point that has been addressed with additional new experiments that assessed the overall expression levels of Mettl3 in TAC hearts- see also our response to your point#6. Mettl3 RNA and protein levels decrease 2-day post TAC but are slightly increased 2 weeks.

These data are now included in the manuscript and mentioned in the text as delineated below.

Page 6, line 198:

This was associated with a decrease in Mettl3 expression 2-day post TAC surgery (Ext.Fig 3C).

8. *Please correct dilative to dilated in page 3.*
9. *There seems a color mismatch for Nde1 in Ext. Fig. 1. Please correct.*
10. *Ext fig 1D is not mentioned in the manuscript*
11. *On page 6, line 179, Extended Fig. 2A was mislabeled instead it should be 3A.*

We have extensively reviewed the manuscript and made sure that the manuscript does not contain the mentioned errors.

12. *Can the authors explain the methods of bioinformatics analysis of m⁶A-seq in more detail?*

Following the reviewer suggestion, we added additional details to the bioinformatic analysis to the method section and mentioned in the text as delineated below

Page 11, line 379:

All sequencing data were subjected to the same preprocessing steps: Reads are trimmed (adapter removal) and quality clipped with Flexbar (Dodt *et al*, 2012). All remaining reads (> 18 bp in length) are mapped either against the murine 45S rRNA precursor sequence (BK000964.3) or the human 45S rRNA precursor sequence (NR_046235.1) with Bowtie 2 to remove rRNA contaminant reads. We employ the splice-aware STAR read aligner (Dobin *et al*, 2013) (release 2.5.1b) to map reads from input RNA and m⁶A IP libraries against the respective Ensembl genome assemblies (release 87). Differential enrichment analyses on IP/input gene read counts was conducted using the edgeR package (McCarthy *et al*, 2012) for each condition separately. We employed a 2-factor design (IP or

input, sample number) and assessed the significance of the IP/input contrast. The adjusted p-values were reported for every expressed gene locus (>10 counts over all input samples).

We are thankful for the constructive critique of reviewer #2 and hopeful that the articulated concerns have been satisfactorily redressed.

Reviewer #3: *Overall, this is an interesting and timely manuscript. The field of epitranscriptomics is growing quickly, and little is known about the role of m6A in the heart. The Authors find that RNA m6A levels are increased in failing human hearts, and that the distribution of m6A different in failing and non-failing samples. They go on to use NRVMs, HL1, and mice to focus on how METTL3 expression regulates m6A levels on cardiac RNA, and how this relates to relevant cardiac phenotypes. Their data show that METTL3 knockdown (loss of m6A) leads to greater cardiac hypertrophy, while METTL3 overexpression (increase of m6A) reduces pathologic cardiomyocyte hypertrophy and fibrosis. Interestingly, cardiac m6A levels are reduced in the acute phase of TAC stress (2d), which is the opposite of what was seen in failing human hearts.*

We thank the reviewer for these supportive remarks and have addressed comments as follows:

Major Critique

In an effort to assess how METTL3 might be regulating these phenotypes, the authors use Ribo-seq in both the animal model (TAC) and cells (HL1). The results here are somewhat confusing and need to be clarified. I think the authors are trying to show that m6A-enriched mRNA transcripts have a higher ribosome occupancy, suggesting higher rates of translation, while mRNA levels are unchanged (Fig 3B and 3E). However, polysome profiling was no different between METTL3 overexpression and control (Fig 3F). Select transcripts did have METTL3-dependent (and presumably m6A-dependent) association with polysomes (Fig 3G). At this point in the manuscript, it is not even clear what material is being assayed (HL1 cells vs mouse hearts; most likely HL1 cells). Finally, the Authors show that there are differences in RNA stability in the control and METTL3-overexpression states, for the same transcripts that show differential polysome loading. Since the Authors have identified Arhgef3 and Myl2 for further study as exemplary genes being regulated by m6A, I suggest they show the following: quantitative Ribo-seq, RNA-seq, and m6A-RIP data for these transcripts, and immunoblotting to demonstrate their protein expression under the control and experimental condition. Furthermore, the Authors should discuss how the results of their data explain the expression of these two genes, and how this relates to m6A-dependent regulation of cardiomyocyte hypertrophy or cardiac fibrosis. Alternative genes could be chosen for this analysis, but the important point is to provide deeper evidence that m6A regulates expression of at least one relevant gene in the heart. For example: Myl2 mRNA seems to be stabilized by m6A, and there is more Myl2 mRNA associated with polysomes. Is there actually more Myl2 protein? If so, is it because there is more RNA, or more efficient translation, or both? Then a discussion of how increased Myl2 expression fits into the overall picture of m6A and the cardiomyocyte hypertrophic response.

This is an excellent point and a very detailed, helpful suggestion to show that m⁶A levels regulate expression of specific transcripts in myocytes. We followed the reviewer's suggestions, reanalyzed data, performed additional m⁶A-RIP PCRs and immunoblots to show that Mettl3 regulates expression of Arhgef3 and Myl2 in myocytes. Mettl3 overexpression does not affect transcript level of Myl2 or Arhgef3 measured by RT-PCR, but increased Myl2 protein expression, whereas Arhgef3 expression was decreased. In line, m⁶A-RIP PCRs confirmed increased enrichment of Arhgef3 and Myl2 transcripts after Mettl3 overexpression. Overall, these new results together with the previous mRNA stability data suggest that m⁶A dependent stabilization of Myl2 transcript results in increased translational efficiency, whereas Arhgef3 transcript stability is decreased and protein expression is reduced after Mettl3 overexpression.

Following suggestion from reviewer#2, we assessed Myl2 and Arhgef3 expression in hypertrophied mouse hearts. Surprisingly, Mettl3 overexpression *in vivo* strongly **reduced** Myl2 expression. We can only speculate that longer Mettl3 overexpression results in different methylation patterns *in vivo*

compared to our *in vitro* data. Unfortunately, we don't have heart lysates after Mettl3 overexpression at earlier timepoints after TAC to analyze the acute response of Mettl3 overexpression on target protein levels.

These data are now included in the manuscript and mentioned in the text as delineated below.

Page 7, line 229

We validated Mettl3-dependent translation of two candidates (*Arhgef3* and *Myl2*) by qRT-PCR on polysomal fractions from control or Mettl3 overexpressing cardiomyocytes. Overall translation was unchanged in Mettl3 overexpressing myocytes as assessed by polysome profiles (Fig 4A). However, *Arhgef3* was significantly less while *Myl2* more translated in our HL-1 Ribo-seq data. In contrast, mRNA levels remain unchanged after Mettl3 overexpression (Fig. 4B). As predicted by our Ribo-seq data, *Arhgef3* transcript levels were decreased in polysomal fractions, whereas *Myl2* levels increased after Mettl3 overexpression (Fig 4C). Increased Mettl3 dependent methylation of *Arhgef3* and *Myl2* transcripts were validated by qRT-PCR after m⁶A-precipitated mRNA from Mettl3- or control-HL-1 cells compared to the input mRNA (Fig.4D).

Page 7, line 242

Finally, Mettl3 overexpression caused decreased *Arhgef3* protein levels, whereas *Myl2* levels were increased in Mettl3 overexpressing myocytes (Fig. 4F-G). In contrast to our *in vitro* data, mice overexpressing Mettl3 in the heart showed decreased *Myl2* expression both during sham and TAC conditions. We speculate that longer Mettl3 overexpression results in different methylation patterns *in vivo* compared to our *in vitro* data. Moreover, different expression levels of m⁶A reader proteins could certainly affect translation of methylated transcripts after long-term overexpression of Mettl3. Ultimately, more studies are needed to fully understand how m⁶A methylation regulates translation in myocytes.

We also added discussion points how this finding could fit into the overall picture of m⁶A dependent gene expression regulation and the hypertrophic growth response.

Page 9, line 286

We validated Mettl3-dependent gene expression regulation of two interesting candidates in follow up experiments. Mettl3 decreases *Arhgef3* protein levels *in vitro*. *Arhgef3* (also known as *Xp1n*) is a Rho guanine nucleotide exchange factor and has been found to interact with the protein kinase mechanistic target of rapamycin (mTOR) (Khanna *et al*, 2013). Increased levels of *Arhgef3* have been shown to stimulate mTORC1. Moreover, increased activity of mTORC1 has been shown to contribute to cardiac hypertrophy and heart failure (Volkers *et al*, 2013; Sciarretta *et al*, 2018), and ongoing studies will investigate the role of *Arhgef3* on mTORC1 regulation during pathological growth. Myosin light chain-2 (*Myl2*) expression increased after Mettl3 overexpression *in vitro* and its expression is slightly increased after TAC surgery. *Myl2* is a sarcomeric protein that belongs to the EF-hand calcium binding protein superfamily. Genetic loss-of-function studies in mice demonstrated the essential role for *Myl2* in cardiac contractile function (Sheikh *et al*, 2015), but it is unknown if increased expression of *Myl2* is needed in addition to the characterized regulation by phosphorylation for the adaptation to increased workload in response to acute pressure overload. Additional studies will be needed to fully understand whether m⁶A dependent gene expression control of *Arhgef3* of *Myl2* levels causally contributes to inhibition of pathological growth after Mettl3 overexpression.

Other Issues

Beyond this major issue, there are a surprisingly large number of other issues that suggest a lack of attention to detail in reporting the findings. The following issues should be also addressed:

We apologize for any confusion and we have extensively reviewed the manuscript and made sure that the manuscript does not contain the mentioned errors.

1. *Line 73: The authors state that it is "conceivable that m6A regulates translation". This is well-established in many publications, including Ref #11. Please revise the manuscript text to reflect the current state of knowledge and cite additional references.*

We apologize for the oversights and missing citations. As a consequence, we have revised the manuscript to include comprehensive information about the current state of knowledge throughout the manuscript.

2. *Ref #17 and #18 are same. Please consolidate these.*

Corrected

3. *ExtFig 1B, IP fraction: Both bars are red; I assume this is an error. Please correct.*

The reviewer is correct, and we have fixed the bar color. See also Reviewer #2 Point 9.

4. *Figure 1A and 1B: Since the sample size is very small (n= 2 control, and n=3 DCM), please use dot plots to show each data point. Also, error bars must be defined for ALL figures (SEM, SD, ?). "*" is used to denote significance, but this is not described in the Figure Legend (?p<0.05). Please ensure that ALL figures denote the meaning of such symbols. Finally, since these are human tissue samples, please include how they were obtained, and cite IRB approval for this.*

We concur with the reviewer. A similar point has been made by Reviewer #2. Accordingly, we have now incorporated RNA-seq results from heart biopsies from a DCM cohort (n=33) compared to non-failing hearts (n=25). Moreover, we have repeated and increased the sample size for the m⁶A measurements in human heart tissues. We also now report dot plots to show each data point. We carefully re-evaluated all figures legends and describe now significance levels for each symbol.

The project utilized human samples from the „Heidelberg Cardio Biobank“ (HCB). The HCB contains tissues such as myocardial biopsies not used for clinical diagnosis and tissue samples obtained from cardiac transplantations. All human samples are pseudonymized according to current requirements of data safety and data protection. A data safety concept approved by the local data safety officer is in place. All samples were collected, processed and stored using state of the art Standard Operating Procedures. In the year 2011 a “broad consent” was granted to the HCB by the local ethics committee which allows the use of human samples for cardiovascular research (S-390/2011). Control RNA from healthy myocardium was commercially obtained from BioCat (Heidelberg) or from tissue that was not used for transplantation for technical reasons which was stored in the HCB.

The new data and information's are now included in the manuscript and mentioned in the text as delineated below.

Page 10, line 323

The characterization of samples and patient data has been approved by the ethics committee, medical faculty of Heidelberg, participants have given written informed consent. Biopsy specimens were obtained from the apical part of the free left ventricular wall (LV) from DCM or control LV biopsy specimens were obtained from stable and symptom-free patients after heart transplantation. Biopsies were rinsed with NaCl (0.9%) and immediately transferred and stored in liquid nitrogen until DNA or RNA was extracted. Total RNA was extracted from biopsies using the RNeasy kit according to the manufacturer's protocol (Qiagen, Germany). For m⁶A-seq analysis, RNA has been isolated from explanted human hearts from patients suffering chronic heart failure. Control RNA from non-failing left ventricular tissue has been obtained commercially (BioCat) or from donor hearts that could not be used for transplantations.

5. *Line 100: Authors state that "thousands of genes were significantly enriched in the IP fraction from failing myocardium". Does this refer to the ~2400 genes shown in Table 1 and Fig 1C? In the Methods, the Authors define enrichment as log FC>0 (and FDR <0.05). Applying these parameters to Table 1 DCM samples identifies 2480 genes as "enriched" in the m6A IP*

sample compared to the control Ab. Notably, this number differs somewhat from the number of "m6A enriched genes in DCM" shown in Fig 1C (1518+877=2395). A similar difference is noted for the Ctrl samples (1216 "enriched" genes applying the stated statistical criteria to Table 1, and 304+877= 1181 genes shown in Fig 1C). Please clarify how exactly how many genes were identified as "m6A enriched" in the human samples, and reconcile Table 1 with Fig 1C.

We apologize for any confusion. We carefully re-evaluated the number enriched genes and corrected the numbers in Fig. 1C using our defined enrichment parameters ($\log_{2}FC > 0$, $FDR < 0.05$). This identified (as correctly stated by the reviewer) 2480 enriched genes in DCM samples and 1216 enriched genes in the Control cohort. Numbers in the Venn diagram now match these numbers (1595+885=2480 and 885 + 331 = 1216). Please note that Enrichment analysis have been performed using the correct numbers from table 1. The corrected diagram is now shown in the revised panel Fig.1C

6. Tables 1-4 are included and referenced but there are no Legends to describe them. Please provide Legends and label the Table headers to indicate which samples are m6A Ab IP vs the ctrl IP.

We have included legends to Table 1-4 and re-labeled sample identifications as suggested by the reviewer.

7. Fig 1F: Color scheme used here is opposite of that in Fig 1A, 1B, and 1C; this is very confusing. Please use a consistent scheme throughout (e.g., blue = ctrl, red = DCM).

We have corrected the color scheme (blue = ctrl, red = DCM).

8. Ext.Fig 1D is not referenced in the text. Please delete this or reference it in the text.

See also Reviewer #2 Point 10. We have added the reference in the revised manuscript as follows

Page 4, line 120:

Specifically, m⁶A-containing transcripts were enriched for genes involved in beta catenin and calmodulin binding (Ext.Fig 1E).

9. ExtFig 2: Please show scale bars for photomicrographs. ExtFig 2C: Two panels are shown, but not labeled. Which one is METTL3 vs inactive mutant?

We have added the scale bars and labeled the panels.

10. Fig 2A: Please explain why the catalytically-dead METTL3 overexpression increases m6A levels. The text of line 154 is not a true statement given the data presented (dead METTL3 did change m6A levels, but it did not alter growth response).

An excellent point that has been also raised by by Reviewer #1 and #2. Two mutations were induced within the catalytic region (CMII) of Mettl3 in order to change amino acid structure. Mutation 1 changed asparagine (AA 395) to Alanine, mutation 2 changes Tryptophan (AA 398) to Alanine, resulting in a mutant Mettl3 lacking methyltransferase activity (aa395-398, DPPW → APPA). This has been shown in previous publications to completely abolish Mettl3 enzymatic activity (Vu *et al*, 2017; Alarcón *et al*, 2015). We cannot rule out that in our cellular system the Mettl3 mutant exhibits partial activity. Since we have repeated the m⁶A measurement, we have also included additional measurement of m⁶A levels in mRNAs from isolated cardiac myocyte after overexpression of mutant Mettl3.

Page 5, line 144:

Overexpression of the enzymatically active Mettl3 significantly increased m⁶A levels, whereas an enzymatically dead mutant of Mettl3 (Vu *et al*, 2017; Alarcón *et al*, 2015) did not increase m⁶A levels

compared to the control group (Fig 2A).

Page 13, line 451:

Generation of the mutant Mettl3 (Mettl3 Mut) construct.

2 point mutations were introduced within the catalytic site of Mettl3 (Vu *et al*, 2017) in two mutagenesis steps by using the following PCR protocol with pAd_Mettl3 as a parent plasmid and specifically designed primers containing the desired mutations using the Agilent QuikChange II Site-Directed Mutagenesis Kit per manufacturers instruction. The obtained plasmid was transformed into NEB alpha-5 E. Coli following the High Efficiency Transformation Protocol from New England Biolabs. Validation of the created plasmid was performed by sequencing and the mutant construct was cloned into a pAd vector using the LR Recombination protocol by Thermo Fisher Scientific.

11. Fig 2B: Why does METTL3 overexpression not reduce cell size, as it does in the heart (Fig 1F)? In Fig 2B, the bar graphs for AdCo, AdMe, and AdMut look identical, including their error bars. The Y-axis is labeled as "relative CSA n-fold change vs AdCo". Please provide the raw data for review.

We now provide raw data about cell size measurements in a new Table for review (Table 5). Additionally, we now provide the data as dot plots instead of bar graphs. We also want to mention that Mettl3 overexpression did reduce cell growth in response to neurohumoral stimulation with phenylephrine (PE) compared to the control group. A similar finding was observed *in vivo*, where Mettl3 overexpression did not affect heart weight after Sham operation but reduced the increase in heart weight after pressure overload (TAC).

The new Figures are now presented in the revised manuscript.

12. Fig 2C-I: What was the AAV "control" used? If it was not the catalytically dead METTL3, why not (given that this was used in Fig 2A and B).

We appreciate the opportunity to address this excellent point. The gene encoding firefly luciferase was cloned into the same AAV9 vector to generate a control AAV9 construct (AAV9-Control), and either AAV9-Mettl3 or AAV9-Control were injected into mice 3 weeks before sham or TAC surgery (Lehmann *et al*, 2017). We did not use the catalytically dead Mettl3, in part because it might exhibit unexpected catalytical activity (see our response to your point 10).

13. Fig 2E: There are four panels; are the right-hand panels +TAC?

Yes, the right-hand panels are the TAC samples. We apologize for mis-labeling and included now the correct labels.

14. ExtFig 2H is not mentioned in the text. It should be referenced on line 144.

We have added the reference in the revised manuscript as follows:

Page 5, line 158

Heart size two weeks after TAC was increased significantly in control mice (Fig 2C-F) and molecular markers of hypertrophy such as *Nppa* and *Nppb* were significantly induced (Ext.Fig 2I).

15. Line 167: "Surprisingly, 50% less overall m6A levels were measured...." The Figure shown does not support this statement. Please state the measured value.

In order to also address reviewer #2 comments, we have performed additional m⁶A measurement (now n=5 for Sham and n=6 for TAC). These results now which are now incorporated into the revised Ext. Fig. 3A and in the text as follows.

Page 6, line 195

Surprisingly, the percentage of m⁶A in mRNA decreased substantially from 0.26% in sham operated animals to 0.12% in response to a TAC surgery (Ext.Fig 3A).

16. Ext Fig 3B and Table 2: Again, there is a discrepancy between the Figure and the Table. Based on the criteria stated in the methods ($\log_{2}FC > 0$ and $FDR < 0.05$), the Table indicates 1567 "m⁶A enriched" genes in the sham set, but the Figure and Text refer to 1543 genes. Table shows 330 "m⁶A enriched" genes in the TAC samples, but the Text and Figure note $206 + 155 = 361$. Please explain/clarify.

Please see also our response to point 5. We carefully re-evaluated the number enriched genes using our defined enrichment parameters ($\log_{2}FC > 0$, $FDR < 0.05$). This identified 1567 enriched genes in Sham hearts and 331 enriched genes in the TAC operated mice. Numbers in the Venn diagram now match these numbers. The corrected diagram is now shown in the revised panel Ext. Fig.3 B.

17. Fig 3B: Please clarify how "Ribo-seq log₂-Fold change" is calculated (normalized Ribo-seq reads per transcript, sham/TAC?). Please also clarify how why the X-axis is showing both Sham and TAC treatments, while the Legend states that each box plot represents Sham/TAC fold-change. Perhaps the left box plot is for "non-m⁶A enriched genes" and the right is for "m⁶A enriched genes"?

For the statistical analyses, we follow the protocol as outlined in Schäfer et al. (Schäfer *et al*, 2015). We pinpoint translational control by identifying significantly differential read counts in Ribo-seq data across conditions and categorize differential gene expression events. We used the edgeR package (McCarthy *et al*, 2012) with a four-factor design matrix (RNA-seq cond1, RNA-seq cond2, Ribo-seq cond1 and Ribo-seq cond2) to accomplish this task. We only consider data points with read count observations across all replicates. The Ribo-seq log₂ Fold change are counts per million reads in TAC vs Sham operated mice.

In the box plot in Fig 3B we calculated the log FC in Ribo-seq (left panel) and RNA-seq (right panel) of enriched methylated transcripts only in Sham conditions (n=1358- Ext. Fig. 3B) in the left plot or enriched methylated transcripts only during TAC (n=122). We re-labeled the graph to clarify this important point.

We updated the method section as follows:

Page 13, line 423

For the analyses, we follow the protocol as outlined in Schäfer et al. (Schäfer *et al*, 2015). We pinpointed translational control by identifying read counts in Ribo-seq data across conditions and categorize differential gene expression events. We use the edgeR package (McCarthy *et al*, 2012) with a four-factor design matrix (RNA-seq cond1, RNA-seq cond2, Ribo-seq cond1 and Ribo-seq cond2) to accomplish this task. We only considered data points with read count observations across all replicates. The Ribo-seq log₂ Fold change are counts per million reads in TAC vs Sham operated mice.

18. Ext Fig 3D: Left panel is not labeled; presumably Ribo-seq data.

The reviewer is correct- we have fixed the missing labeling

19. Line 187: Here, the Authors introduce a new model: HL1 cells. This needs to be clearly stated in the text of the manuscript.

Important point. We have added information's about the new model in the revised manuscript as follows:

Page 7, line 215:

Since Ribo-Seq methods still require a large number of cells, we used the murine HL-1 cardiomyocyte cell line as our source material. HL-1 cells can be used as a cardiomyocyte model as they have key characteristics of cardiac myocytes, although their metabolism and structure are less

organized compared to primary cardiac myocytes (Claycomb *et al*, 1998; Eimre *et al*, 2008). Confluent and spontaneously beating murine HL-1 cells were infected with Mettl3 adenovirus to increase m⁶A-levels (Ext.Fig 3G). We identified Mettl3-dependent, highly differentially translated mRNAs by Ribo-seq (Fig 3C, Ext. Fig.3H and Table 3).

Page 13, line 443

HL-1 cell culture:

HL-1 cardiomyocytes were maintained as described (Claycomb *et al*, 1998).

20. Line 196: Table 1 is referenced here; I presume this should actually be Table 3.

The reviewer is correct- we have fixed the reference as follows:

Page 7, line 220

We identified Mettl3-dependent, highly differentially translated mRNAs by Ribo-seq (Fig 3C, Ext. Fig.3H and Table 3).

21. Other than the criticisms listed here, the Authors should be commended for a thorough and detailed Methods section.

Reference:

- Alarcón CR, Lee H, Goodarzi H, Halberg N & Tavazoie SF (2015) N⁶-methyladenosine marks primary microRNAs for processing. *Nature* **519**: 482–485 Available at: <http://www.nature.com/articles/nature14281> [Accessed January 29, 2019]
- Claycomb WC, Lanson NA, Stallworth BS, Egeland DB, Delcarpio JB, Bahinski A & Izzo NJ (1998) HL-1 cells: A cardiac muscle cell line that contracts and retains phenotypic characteristics of the adult cardiomyocyte. *Proc. Natl. Acad. Sci.* **95**: 2979–2984 Available at: <http://www.pnas.org/cgi/doi/10.1073/pnas.95.6.2979> [Accessed February 1, 2019]
- Dobin A, Davis CA, Schlesinger F, Drenkow J, Zaleski C, Jha S, Batut P, Chaisson M & Gingeras TR (2013) STAR: ultrafast universal RNA-seq aligner - Supplementary Data. *Bioinformatics* **29**: 15–21 Available at: <http://bioinformatics.oxfordjournals.org/content/suppl/2012/10/25/bts635.DC1/Dobin.STAR.Supplementary.Response2.pdf>
- Dotz M, Roehr J, Ahmed R & Dieterich C (2012) FLEXBAR—Flexible Barcode and Adapter Processing for Next-Generation Sequencing Platforms. *Biology (Basel)*. **1**: 895–905 Available at: <http://www.mdpi.com/2079-7737/1/3/895/>
- Dorn LE, Lasman L, Chen J, Xu X, Berlo JH Van & Accornero F (2018) The m⁶A mRNA Methylase

- METTL3 Controls Cardiac Homeostasis and Hypertrophy. *Circulation* **139**: 533–545 Available at: <http://www.ncbi.nlm.nih.gov/pubmed/30586742> [Accessed January 31, 2019]
- Eimre M, Paju K, Pelloux S, Beraud N, Roosimaa M, Kadaja L, Gruno M, Peet N, Orlova E, Remmelkoor R, Piirsoo A, Saks V & Seppet E (2008) Distinct organization of energy metabolism in HL-1 cardiac cell line and cardiomyocytes. *Biochim. Biophys. Acta - Bioenerg.* **1777**: 514–524 Available at: <https://www.sciencedirect.com/science/article/pii/S0005272808000625> [Accessed February 1, 2019]
- He C (2010) Grand Challenge Commentary: RNA epigenetics? *Nat. Chem. Biol.* **6**: 863–865 Available at: <http://www.nature.com/doi/10.1038/nchembio.482> [Accessed January 31, 2019]
- Jia G, Fu Y, Zhao X, Dai Q, Zheng G, Yang Y, Yi C, Lindahl T, Pan T, Yang YG & He C (2011) N6-Methyladenosine in nuclear RNA is a major substrate of the obesity-associated FTO. *Nat. Chem. Biol.* **7**: 885–887
- Khanna N, Fang Y, Yoon M-S & Chen J (2013) XPLN is an endogenous inhibitor of mTORC2. *Proc. Natl. Acad. Sci. U. S. A.* **110**: 15979–84 Available at: <http://www.ncbi.nlm.nih.gov/pubmed/24043828> [Accessed February 4, 2019]
- Lehmann LH, Jebessa ZH, Kreuzer MM, Horsch A, He T, Kronlage M, Dewenter M, Sramek V, Oehl U, Krebs-Haupenthal J, von der Lieth AH, Schmidt A, Sun Q, Ritterhoff J, Finke D, Völkers M, Jungmann A, Sauer SW, Thiel C, Nickel A, et al (2017) A proteolytic fragment of histone deacetylase 4 protects the heart from failure by regulating the hexosamine biosynthetic pathway. *Nat. Med.* **24**: 62–72 Available at: <http://www.nature.com/doi/10.1038/nm.4452> [Accessed January 29, 2019]
- Mathiyalagan P, Adamiak M, Mayourian J, Sassi Y, Liang Y, Agarwal N, Jha D, Zhang S, Kohlbrenner E, Chepurko E, Chen J, Trivieri MG, Singh R, Bouchareb R, Fish K, Ishikawa K, Lebeche D, Hajjar RJ & Sahoo S (2018) FTO-Dependent m6A Regulates Cardiac Function During Remodeling and Repair. *Circulation*: CIRCULATIONAHA.118.033794 Available at: <http://circ.ahajournals.org/lookup/doi/10.1161/CIRCULATIONAHA.118.033794>
- Mauer J, Luo X, Blanjoie A, Jiao X, Grozhik A V., Patil DP, Linder B, Pickering BF, Vasseur JJ, Chen Q, Gross SS, Elemento O, Debart F, Kiledjian M & Jaffrey SR (2017) Reversible methylation of m6Amin the 5' cap controls mRNA stability. *Nature* **541**: 371–375
- McCarthy DJ, Chen Y & Smyth GK (2012) Differential expression analysis of multifactor RNA-Seq experiments with respect to biological variation. *Nucleic Acids Res.* **40**: 4288–4297
- Meder B, Haas J, Sedaghat-Hamedani F, Kayvanpour E, Frese K, Lai A, Nietsch R, Scheiner C, Mester S, Bordalo DM, Amr A, Dietrich C, Pils D, Siede D, Hund H, Bauer A, Holzer DB, Ruhparwar A, Mueller-Hennessen M, Weichenhan D, et al (2017) Epigenome-Wide Association Study Identifies Cardiac Gene Patterning and a Novel Class of Biomarkers for Heart Failure. *Circulation* **136**: 1528–1544 Available at: <http://circ.ahajournals.org/lookup/doi/10.1161/CIRCULATIONAHA.117.027355> [Accessed February 5, 2019]
- Schafer S, Adami E, Heinig M, Rodrigues KEC, Kreuchwig F, Silhavy J, Van Heesch S, Simate D, Rajewsky N, Cuppen E, Pravenec M, Vingron M, Cook SA & Hubner N (2015) Translational regulation shapes the molecular landscape of complex disease phenotypes. *Nat. Commun.* **6**: 7200
- Sciarretta S, Forte M, Frati G & Sadoshima J (2018) New insights into the role of mtor signaling in the cardiovascular system. *Circ. Res.* **122**: 489–505 Available at: <http://www.ncbi.nlm.nih.gov/pubmed/29420210> [Accessed November 21, 2018]
- Sheikh F, Lyon RC & Chen J (2015) Functions of myosin light chain-2 (MYL2) in cardiac muscle and disease. *Gene* **569**: 14–20 Available at: <https://www.sciencedirect.com/science/article/pii/S0378111915007350?via%3Dihub> [Accessed February 4, 2019]
- Shi H, Wang X, Lu Z, Zhao BS, Ma H, Hsu PJ, Liu C & He C (2017) YTHDF3 facilitates translation and decay of N6-methyladenosine-modified RNA. *Cell Res.* **27**: 315–328 Available at: <http://www.ncbi.nlm.nih.gov/pubmed/28106072> [Accessed February 4, 2019]
- Shi H, Zhang X, Weng YL, Lu Z, Liu Y, Lu Z, Li J, Hao P, Zhang Y, Zhang F, Wu Y, Delgado JY, Su Y, Patel MJ, Cao X, Shen B, Huang X, Ming G li, Zhuang X, Song H, et al (2018) m6A facilitates hippocampus-dependent learning and memory through YTHDF1. *Nature* **563**: 249–253 Available at: <http://www.ncbi.nlm.nih.gov/pubmed/30401835> [Accessed February 4, 2019]
- Volkers M, Toko H, Doroudgar S, Din S, Quijada P, Joyo AY, Ornelas L, Joyo E, Thuerauf DJ, Konstandin MH, Gude N, Glembotski CC & Sussman MA (2013) Pathological hypertrophy amelioration by PRAS40-mediated inhibition of mTORC1. *Proc. Natl. Acad. Sci.* **110**: 12661–12666 Available at: <http://www.pnas.org/cgi/doi/10.1073/pnas.1301455110>
- Vu LP, Pickering BF, Cheng Y, Zaccara S, Nguyen D, Minuesa G, Chou T, Chow A, Saletore Y, Mackay M, Schulman J, Famulare C, Patel M, Klimek VM, Garrett-Bakelman FE, Melnick A, Carroll M, Mason CE, Jaffrey SR & Kharas MG (2017) The N6-methyladenosine (m6A)-forming enzyme METTL3 controls myeloid differentiation of normal hematopoietic and leukemia cells. *Nat. Med.* **23**: 1369–1376 Available at: <http://www.ncbi.nlm.nih.gov/pubmed/28920958> [Accessed January 29, 2019]
- Wang X, Zhao BS, Roundtree IA, Lu Z, Han D, Ma H, Weng X, Chen K, Shi H & He C (2015) N6-methyladenosine modulates messenger RNA translation efficiency. *Cell* **161**: 1388–1399

Wei J, Liu F, Lu Z, Fei Q, Ai Y, He PC, Shi H, Cui X, Su R, Klungland A, Jia G, Chen J & He C (2018)
Differential m6A, m6Am, and m1A Demethylation Mediated by FTO in the Cell Nucleus and Cytoplasm.
Mol. Cell **71**: 973–985.e5 Available at: <https://doi.org/10.1016/j.molcel.2018.08.011> [Accessed January
31, 2019]

February 26, 2019

RE: Life Science Alliance Manuscript #LSA-2018-00233-TR

Dr. Mirko Völkers
Heidelberg University Hospital
Internal Medicine, Cardiology
Im Neuenheimer Feld 410
Heidelberg, 69120
Germany

Dear Dr. Völkers,

Thank you for submitting your revised manuscript entitled "m6A-mRNA methylation regulates cardiac gene expression and cellular growth". As you will see, the reviewers appreciate the work added in revision, but think that a few issues, some of which pertain to the new data added, still need to get addressed. We would thus like to invite you to further revise your manuscript, addressing the comments of the reviewers. Additionally, the following revisions are necessary to meet our formatting guidelines:

- please provide the manuscript text as a word doc file and list all contributing authors
- please upload all figures as individual files, including the supplementary figures
- please note that we have Supplementary figures (not Ext figures) at Life Science Alliance
- please add a callout to Fig 4H and to table 4 and table 5 in the text
- please add the statistical test used in figure 4 to the legend

A. FINAL FILES:

-- Summary blurb (enter in submission system): A short text summarizing in a single sentence the study (max. 200 characters including spaces). This text is used in conjunction with the titles of papers, hence should be informative and complementary to the title. It should describe the context

and significance of the findings for a general readership; it should be written in the present tense and refer to the work in the third person. Author names should not be mentioned.

B. MANUSCRIPT ORGANIZATION AND FORMATTING:

Sincerely,

Reviewer #1 (Comments to the Authors (Required)):

The new Figure 1A and Figure 1B worries me now. The m6A quantification shown in 1A might not be accurate, or there are other mechanisms operating in this system.

Reviewer #2 (Comments to the Authors (Required)):

The authors have addressed all of my comments with adequate in vitro and in vivo data as well as careful reevaluation of text. I have one major suggestion to the authors and I invite authors to respond to my suggestion before having this paper accepted for publication.

In the previous version of manuscript, they showed increased METTL3 and decreased FTO expression (also for WTAP, METTL14). Now from a published dataset, they report no changes to METTL3 and FTO and other m6A regulators. This is contradictory. While there is a major difference in "n" numbers between their data and published RNA-seq analysis, they should present both in the manuscript i.e. results from their own experiments showing differences in METTL3 and other m6a regulators and also analysis from RNA-seq datasets showing no changes. This will help the m6A cardiac field to better understand molecular mechanisms.

Reviewer #3 (Comments to the Authors (Required)):

Overall, the authors have been diligent in addressing reviewer concerns, and I do feel that the manuscript has been significantly improved. However, there remain a few issues that should be addressed.

In lines 150-151, the authors state, "inactive Mettl3 mutant did not block cell growth in response to PE, but rather augmented cellular size compared to control cells". However, the data does not support this statement. The data in Fig 2B show that the inactive Mettl3 expression had no effect on cellular hypertrophy, while the active Mettl3 blocked cell growth. Therefore, the inactive Mettl3 did not "augment" cellular size; instead, it had "no effect".

Please address the effect of Mettl3 overexpression on cardiomyocyte size in Fig 2E-F. It is not discussed in the manuscript. Although the figure indicates statistically significant difference between Co TAC and Mettl3 TAC, the difference seems trivial. How many different animals were analyzed? Each animal is a biological replicate, not each cell. The same criticism applies to Fig 2B, but the differences there seem obvious. Also, it might be visually better to group the sham animals together and the TAC animals together, since the difference we are most interested to see is ctrl vs Mettl3, not sham vs TAC.

Figure 4H is not specifically called out in the manuscript (though it is briefly referenced, line 245-246). Please describe the results of this figure in the text, and also quantify the results in Fig 4H. Arhgef3 protein seems to be decreased in response to Mettl3 overexpression (sham animals; "expected" from cell culture results) but also definitely increased in response to TAC (both control and Mettl3). Myl2 is definitely decreased in the Mettl3 model, which is opposite of what might be expected from the cell culture data. The authors "speculate that longer Mettl3 overexpression results in different methylation patterns in vivo compared to our in vitro data". It should be trivial to

test the m6A status of these transcripts from the in vivo samples, using RT-qPCR to amplify the RNA samples shown in Fig 3B (already subjected to m6A isolation).

Dr. Andrea Leibfried
Executive Editor
Life Science Alliance

Dear Dr. Leibfried

Thank you for inviting us to re-revise our submission “m⁶A-mRNA methylation regulates cardiac gene expression and cellular growth”. We thank the reviewers for their careful analysis of our manuscript and hope you will now find our work acceptable for publication in Life Science Alliance. Our specific responses and references to changes in the revised manuscript are delineated as follows:

Response to Reviewers:

Reviewer #1: *The new Figure 1A and Figure 1B worries me now. The m6A quantification shown in 1A might not be accurate, or there are other mechanisms operating in this system.*

A similar point has been raised by reviewer #2. Following suggestions of reviewer #2, we now present our own data showing differences in METTL3 expression and also analysis from published RNA-seq datasets from a larger DCM cohort in the revised manuscript. We do see changes in transcript and protein levels in Mettl3 in a small cohort of DCM patients. Please note that we performed m⁶A-quantification on those patients, and we see increased overall m⁶A levels in those human heart failure samples in line with increased Mettl3 expression. Inherited DCM is caused by mutations in a variety of individual genes. However, individual disease-causing mutation are often unknown. Unfortunately, we do not have detailed information about the etiology of DCM hearts and hence we cannot explain why only a subset of patients do show changes in Mettl3 expression. Moreover, it could be possible that the failing heart has altered Mettl3 or Fto activity without changes in expression levels due to changes in enzymatic activity. The revised information is now shown in the Figure panels and mentioned in the revised manuscript as follows:

Page 4, line 108

These data did not show any significant changes of Mettl3 or FTO on the transcript level assessed by RNA-seq (Fig 1B). A trend in increased protein levels and RNA levels of Mettl3 could be observed in a small cohort of DCM hearts compared to control samples (Supplementary Fig 1C), suggesting that only in subset of DCM patients Mettl3 expression levels increased.

We thank the reviewer for the erudite commentary and hope these revisions serve to appropriately address the provided critique.

Reviewer #2: *The authors have addressed all of my comments with adequate in vitro and in vivo data as well as careful reevaluation of text. I have one major suggestion to the authors, and I invite authors to respond to my suggestion before having this paper accepted for publication.*

We appreciate the reviewer's assessment.

Major concern:

1. *In the previous version of manuscript, they showed increased METTL3 and decreased FTO expression (also for WTAP, METTL14). Now from a published dataset, they report no changes to METTL3 and FTO and other m6A regulators. This is contradictory. While there is a major difference in "n" numbers between their data and published RNA-seq analysis, they should present both in the manuscript i.e. results from their own experiments showing differences in METTL3 and other m6a regulators and also analysis from RNA-seq datasets showing no changes. This will help the m6A cardiac field to better understand molecular mechanisms.*

The reviewer raises a good point and a similar point has raised by reviewer#1. We agree and followed the suggestions. We now show both RNA-seq data in the revised manuscript and in addition own data with RT-PCR data and immunoblots from a smaller cohort.

The revised information is now shown in the Figure panels and mentioned in the revised manuscript as follows:

Page 4, line 108

These data did not show any significant changes of Mettl3 or FTO on the transcript level assessed by RNA-seq (Fig 1B). A trend in increased protein levels and RNA levels of Mettl3 could be observed in a small cohort of DCM hearts compared to control samples (Supplementary Fig 1C), suggesting that only in subset of DCM patient's Mettl3 expression levels increased.

We are thankful for the constructive critique of reviewer #2 and hopeful that the articulated concerns have been satisfactorily redressed.

Reviewer #3: *Overall, the authors have been diligent in addressing reviewer concerns, and I do feel that the manuscript has been significantly improved. However, there remain a few issues that should be addressed.*

We thank the reviewer for these supportive remarks

Major concern:

1. *In lines 150-151, the authors state, "inactive Mettl3 mutant did not block cell growth in response to PE, but rather augmented cellular size compared to control cells". However, the data does not support this statement. The data in Fig 2B show that the inactive Mettl3 expression had no effect on cellular hypertrophy, while the active Mettl3 blocked cell growth. Therefore, the inactive Mettl3 did not "augment" cellular size; instead, it had "no effect".*

We appreciate the opportunity to clarify this point. Although maybe not clearly visible in the Fig.2b, myocytes after PE treatment expressing inactive Mettl3 mutant are slightly bigger than myocytes treated with the control virus in response to PE treatment (4th bar compared to the last bar), which is significant ($p < 0.048$, Anova with Bonferoni multiple comparison test. Mean size Control + PE = $762 \mu\text{M}^2$, AdMettl3 Mutant +PE = $825 \mu\text{M}^2$). We weakened our statement as follows:

Page 5, line 153

inactive Mettl3 mutant did not block cell growth, but slightly augmented cellular size compared to control cells when treated with PE.

2. *Please address the effect of Mettl3 overexpression on cardiomyocyte size in Fig 2E-F. It is not discussed in the manuscript. Although the figure indicates statistically significant difference between Co TAC and Mettl3 TAC, the difference seems trivial. How many different animals were analyzed? Each animal is a biological replicate, not each cell. The same criticism applies to Fig 2B, but the differences there seem obvious. Also, it might be visually better to group the sham animals together and the TAC animals together, since the difference we are most interested to see is ctrl vs Mettl3, not sham vs TAC.*

We agree that the difference is small. We measured the cell size in 4 individual animals in each group (in total 200-300 cells per group). We followed the reviewer suggestions and now group the sham animals together and the TAC animals together throughout the entire revised Fig.2. We revised the manuscript and mentioned these findings in the revised manuscript as follows:

Page 5, line 165

Pathological hypertrophic cellular growth was attenuated in hearts of Mettl3 overexpressing mice, as evidenced by cross sectional area of myocytes (Fig 2E-F). Myocytes in Mettl3 overexpressing

hearts were significantly enlarged two weeks post TAC surgery compared to sham operated animals, but smaller compared than control TAC mice, without significant differences in hypertrophy marker expression between control TAC and *Mettl3* TAC mice (Supplementary Fig 2I).

Page 21, line

F) Cell surface area measurement from WGA staining (n= 4 animals per group and 200-300 cells in total, * p < 0.05, **** p < 0.0001 by one way Anova).

3. *Figure 4H is not specifically called out in the manuscript (though it is briefly referenced, line 245-246). Please described the results of this figure in the text, and also quantify the results in Fig 4H.*

We apologize for this mistake. We quantified results of Fig.4H and revised the manuscript as follows:

Page 8, line 254

In contrast to our *in vitro* data, mice overexpressing *Mettl3* in the heart showed decreased *Myl2* expression both during sham and TAC conditions (Fig. 4H-I), whereas *Arhgef3* expression both increased during TAC conditions in *Mettl3* overexpressing mice and in control animals.

4. *Arhgef3 protein seems to be decreased in response to Mettl3 overexpression (sham animals; "expected" from cell culture results) but also definitely increased in response to TAC (both control and Mettl3). Myl2 is definitely decreased in the Mettl3 model, which is opposite of what might be expected from the cell culture data. The authors "speculate that longer Mettl3 overexpression results in different methylation patterns in vivo compared to our in vitro data". It should be trivial to test the m6A status of these transcripts from the in vivo samples, using RT-qPCR to amplify the RNA samples shown in Fig 3B (already subjected to m6A isolation).*

The reviewer raises another good point and we agree that these data are important. Following his suggestions, we performed RT-qPCRs to quantify the enrichment of both *Arhgef3* and *Myl2* transcripts after m⁶A-IP after different timepoints after TAC surgery. Please note that we don't have RNA-samples after m⁶A-pulldown from our *in vivo* *Mettl3* overexpression experiments. Thus, we cannot fully answer the question whether longer *Mettl3* overexpression results in different methylation compared to control animals during TAC surgery. Nevertheless, our new data now show that m⁶A enrichment of *Arhgef3* and *Myl2* changed in response to TAC surgery. *Myl2* enrichment after m⁶A pulldown increased early 2-day post TAC surgery, whereas *Arhgef3* enrichment was the highest 2-week post TAC surgery. To test the impact of increased *Mettl3* expression on these methylation dynamics, we would need to perform additional *in vivo* experiments since we don't have enough mRNA material for the m⁶A IP from *Mettl3* overexpressing mice. We revised the manuscript and mentioned these findings in the revised manuscript as follows:

Page 8, line 257

Intriguingly, m⁶A enrichment of *Arhgef3* and *Myl2* changed at different timepoints after TAC compared to sham-operated animals (Supplementary Fig 3I). We speculate that longer *Mettl3* overexpression results in different methylation patterns in vivo compared to our *in vitro* data.

We are thankful for the constructive critique of reviewer #3 and hopeful that the articulated concerns have been now satisfactorily redressed.

Thank you again for fair and reasonable requests for manuscript revisions.

Best regards

March 22, 2019

RE: Life Science Alliance Manuscript #LSA-2018-00233-TRR

Dr. Mirko Völkers
Heidelberg University Hospital
Internal Medicine, Cardiology
Im Neuenheimer Feld 410
Heidelberg, --- Select One --- 69120
Germany

Dear Dr. Völkers,

Thank you for submitting your Research Article entitled "m6A-mRNA methylation regulates cardiac gene expression and cellular growth". I appreciate the introduced changes and it is a pleasure to let you know that your manuscript is now accepted for publication in Life Science Alliance. Congratulations on this interesting work.

DISTRIBUTION OF MATERIALS:

Again, congratulations on a very nice paper. I hope you found the review process to be constructive and are pleased with how the manuscript was handled editorially. We look forward to future exciting submissions from your lab.

Sincerely,
